# Current-induced switching of a van der Waals ferromagnet at room temperature

**Shivam N. Kajale[1,4], Thanh Nguyen[2,4], Corson A. Chao[3], David C. Bono[3], Artittaya Boonkird[2], Mingda Li [2] & Deblina Sarkar [1] ✉**

Recent discovery of emergent magnetism in van der Waals magnetic materials (vdWMM) has broadened the material space for developing spintronic devices for energy-efficient computation. While there has been appreciable progress in vdWMM discovery, a solution for non-volatile, deterministic switching of vdWMMs at room temperature has been missing, limiting the prospects of their adoption into commercial spintronic devices. Here, we report the first demonstration of current-controlled non-volatile, deterministic magnetization switching in a vdW magnetic material at room temperature. We have achieved spin-orbit torque (SOT) switching of the PMA vdW ferromagnet $Fe_3GaTe_2$ using a Pt spin-Hall layer up to 320 K, with a threshold switching current density as low as $J_{sw} = 1.69 \times 10^6$ A cm$^{-2}$ at room temperature. We have also quantitatively estimated the anti-damping-like SOT efficiency of our $Fe_3GaTe_2$/Pt bilayer system to be $\xi_{DL} = 0.093$, using the second harmonic Hall voltage measurement technique. These results mark a crucial step in making vdW magnetic materials a viable choice for the development of scalable, energy-efficient spintronic devices.

Magnetic materials-based spintronic devices[1–3] hold great promise as energy-efficient, non-volatile memories and building blocks of neuromorphic[4] and probabilistic[5,6] computing hardware. However, just a few optimal material systems, like CoFeB/MgO[7–9] in particular, have been identified for scalable device applications over the last two decades. The discovery of emergent magnetism in two-dimensional van der Waals (vdW) magnetic materials[10–13] has opened a new arena for material exploration for spintronic technologies[14]. vdW magnetic materials provide scalable, perpendicular magnetic anisotropy (PMA) alternatives to CoFeB/MgO while also providing atomically smooth interfaces down to monolayer thicknesses to help maintain device performance. However, a crucial requirement for the development of spintronic devices with vdW materials is the deterministic switching of PMA magnetism using current or voltage drives at room temperature.

The $Fe_nGeTe_2$ family of metallic ferromagnets[12,15,16] has shown favorable signs for technological translation in recent years, with Curie temperature reaching close to room temperature,

while also exhibiting a strong PMA[12,15]. Several reports on the current-control of PMA magnetization in $Fe_3GeTe_2$ have appeared, in spin-orbit torque (SOT) systems utilizing heavy metals[17,18], topological insulators[19] and topological semimetals[20–22]. However, these reports are limited to operation only at cryogenic temperatures (up to 200 K). The recently reported vdW ferromagnet, $Fe_3GaTe_2$ (FGaT), which is a close analogue of $Fe_3GeTe_2$, holds promise in this regard[23]. It is a metallic ferromagnet, with Curie temperature exceeding 350 K and a strong perpendicular magnetic anisotropy up to $3.88 \times 10^5$ J m$^{-3}$, making it suitable for developing scalable and stable magnetic tunnel junctions[23]. In fact, FGaT based magnetic tunnels junctions (and spin-valves in general) have already been reported[24–27], with an impressive tunneling magneto-resistance (TMR) ratio up to 85% at 300 K observed in the FGaT/WSe₂/FGaT system[24]. This makes FGaT a favorable candidate for exploring room temperature current-based control of PMA magnetization in vdW material systems.

[1]MIT Media Lab, Massachusetts Institute of Technology, Cambridge, MA, USA. [2]Department of Nuclear Science and Engineering, Massachusetts Institute of Technology, Cambridge, MA, USA. [3]Department of Materials Science and Engineering, Massachusetts Institute of Technology, Cambridge, MA, USA. [4]These authors contributed equally: Shivam N. Kajale, Thanh Nguyen. ✉e-mail: deblina@mit.edu

Here, we report the non-volatile, deterministic magnetization switching at room temperature of the vdW ferromagnet using $Fe_3GaTe_2$/Pt bilayer devices. Bulk crystals of $Fe_3GaTe_2$ were grown using a self-flux method, and its magnetic properties were studied in the bulk and in exfoliated sheets. Bilayer devices of multi-layer FGaT and 6 nm Pt were fabricated to demonstrate type-z spin-orbit torque switching[28,29] of FGaT magnetization up to 320 K in the presence of a 100 Oe in-plane magnetic field, with a threshold current density of $1.69 \times 10^6$ A cm$^{-2}$. Furthermore, second harmonic Hall voltage measurements were used to estimate the anti-damping-like field component which is responsible for the current-induced switching, and the SOT efficiency of the FGaT/Pt system.

## Results

### Growth and characterization of $Fe_3GaTe_2$

We first present our characterization of bulk crystalline vdW FGaT synthesized using a Te self-flux method (see Methods). The FGaT unit cell possesses a hexagonal symmetry with space group $P6_3/mmc$ (no. 194), like that of isostructural $Fe_3GeTe_2$, wherein two adjacent quintuple layer substructures with two inequivalent Fe crystallographic sites form a vdW gap between the tellurium layers (Fig. 1a). Millimeter-sized hexagonal planar crystals were measured with powder X-ray diffraction and showcase prominent (00 L) Bragg peaks which were analyzed using a Rietveld refinement (Fig. 1b). To verify the composition of our crystals, we performed energy-dispersive X-ray

spectroscopy elemental mapping on a diverse set of bulk crystals and exfoliated flakes which all exhibited the correct atomic ratio (Fig. 1c). Hysteresis loops of the direct current magnetization on bulk FGaT were performed from 3 K to 400 K with a magnetic field applied out-of-plane (OOP) as shown in Fig. 1d. Both the coercive field and the saturation magnetization gradually decrease with increasing temperature up to the transition temperature near 340 K. These are complemented by measurements of the Hall resistivity with magnetic field applied out-of-plane which showcase a prominent anomalous Hall effect (AHE) accompanying the magnetization up to the same temperature (Fig. 1e) with a room temperature Hall angle of 2.6°. The temperature dependence of magnetization with an applied out-of-plane magnetic field of 1000 Oe under field-cooling shows a departure near the Curie temperature from the zero-field-cooling situation (Fig. 1f). The heat capacity manifests a prominent peak (Fig. 1g) and the temperature-dependent longitudinal resistivity displays a noticeable change of slope (Fig. 1h) both near 340 K. Altogether, these complementary measurements consistently indicate room-temperature magnetic properties and the high ferromagnetic transition temperature of the bulk FGaT crystals.

Next, we have characterized magnetism in exfoliated nanosheets of FGaT using a combination of AHE and polar magneto-optical Kerr effect (MOKE) measurements. Figure 2a shows the temperature dependent AHE hysteresis loops for OOP field sweep for a 29 nm thick FGaT flake. As depicted in the schematic (Fig. 2a inset), current is

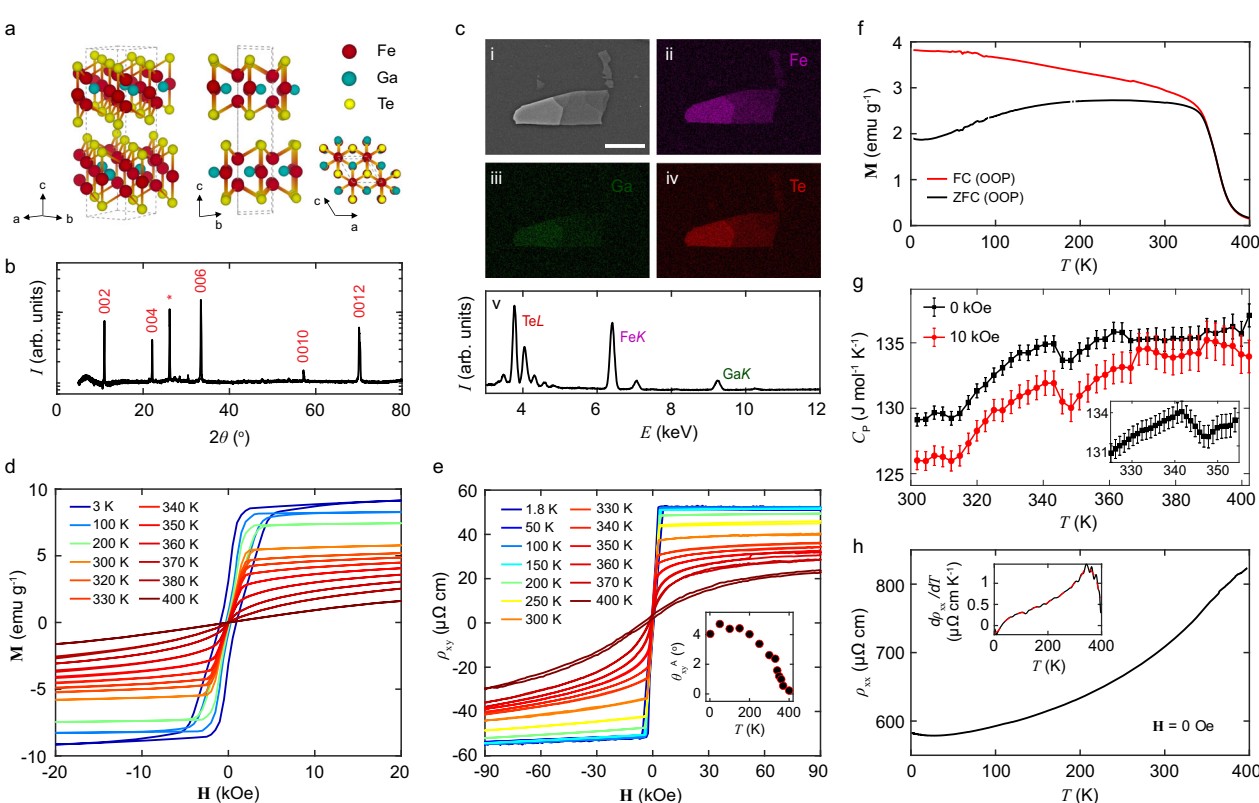

**Fig. 1 | Characterization of $Fe_3GaTe_2$ bulk crystals. a** Crystal structure of $Fe_3GaTe_2$ under different orientations: general view, *bc*-plane and *ac*-plane. **b** Powder x-ray diffraction (PXRD) pattern of the bulk crystal with indexed prominent Bragg peaks. **c** Scanning electron microscope image of an exfoliated flake on $SiO_2$/Si substrate (i) with corresponding elemental Fe (ii), Ga (iii) and Te (iv) maps and energy-dispersive spectra (v). The scale bar on the SEM image is 20 μm. **d** Spontaneous magnetization dependence with magnetic field applied out-of-plane at different temperatures. **e** Hall resistivity of a bulk sample measured at different temperatures with the magnetic field applied in the out-of-plane direction. Inset shows the measured anomalous Hall angle as a function of temperature.

**f** Spontaneous magnetization under field-cooled (FC) and zero-field-cooled (ZFC) conditions with respect to temperature with the magnetic field (1000 Oe) applied out-of-plane. **g** Specific heat capacity measurement with zero and a 10 kOe magnetic field applied in the out-of-plane direction. Inset shows a fine scan at zero field near the magnetic transition showing a small upturn near the magnetic transition temperature. Error bars represent the standard deviation. **h** Temperature dependence of the bulk longitudinal resistivity. Inset shows the derivative of the longitudinal resistivity with respect to temperature. A broad peak indicates the presence of a magnetic transition.

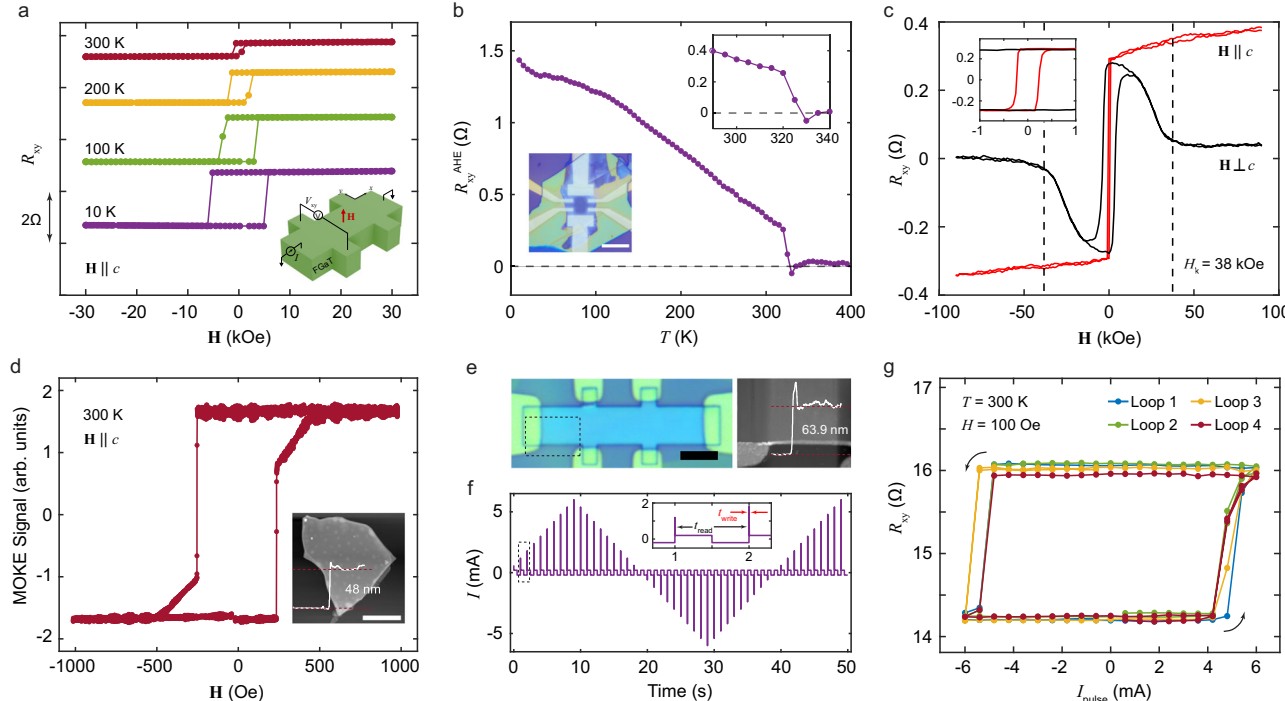

**Fig. 2 | Magneto-transport in exfoliated FGaT and switching of FGaT/Pt devices.** **a** Temperature dependent hysteresis plots of the device for out-of-plane field sweeps (**H** ∥ *c*). Data offset along *y*-axis for clarity. Inset: Schematic of measurement geometry. **b** Variation of anomalous Hall resistance of a (29 nm) FGaT flake against temperature, indicating a Curie temperature of ≈ 328 K. Insets – optical image of the FGaT Hall bar on bottom left, zoomed in view of $R_{xy}^{AHE} - T$ close to the Curie temperature. **c** Comparison of room temperature Hall resistance curves of the FGaT device for field swept OOP (**H** ∥ *c*) and in-plane (**H** ⊥ *c*), with anisotropy field of $H_k$ = 38 kOe denoted by vertical dashed lines. Inset: Low field zoom-in view of the plots. **d** Room temperature polar MOKE curve of a (48 nm) FGaT flake, indicative of strong PMA and a coercivity $H_c$ = 235 Oe. **e** Optical image of the FGaT (57.9 nm)/Pt (6 nm) device (left) and its AFM micrograph (right). **f** Current sequence applied to the FGaT/Pt device for magnetization switching experiments, with the inset clarifying the nature of write pulses ($t_{write}$) and read pulses ($t_{read}$). **g** Cyclic magnetization switching curves observed for the FGaT/Pt devices over four consecutives current pulsing loops, at 300 K under in-plane bias field of 100 Oe. Scale bars: 5 μm.

applied along the *x* – axis and the transverse voltage ($V_{xy}$) is measured along *y* – axis for the AHE measurements. The hysteresis loops exhibit a rectangular nature right up to room temperature, indicating a strong perpendicular magnetic anisotropy. Figure 2b shows the temperature dependence of remanent anomalous Hall resistance. The $R_{xy}^{AHE}$ plotted here is the difference in transverse resistance of the device, recorded while warming up the device at zero magnetic field, after cooling under +3 T and −3 T out-of-plane field, respectively. The sharp drop in $R_{xy}^{AHE}$ above 320 K (inset of Fig. 2b) marks the transition from ferromagnetic to paramagnetic state with a Curie temperature $T_C$ ≈ 328 K, slightly lower than the bulk value. The material exhibits an effective anisotropy field of $H_k$ = 38 kOe at 300 K as evident from the $R_{xy}$ plot for near in-plane magnetic field sweeps (**H** ⊥ *c*) in Fig. 2c. The inset provides $R_{xy}$ vs **H** with finer field steps indicating a room temperature coercivity of 220 Oe for the OOP field sweep. To further confirm room temperature magnetic properties of the FGaT flakes, we report polar MOKE measurement of a 48 nm-thick flake in Fig. 2d. The hysteresis plot shows a near 100% remanence confirming the presence of strong PMA, with a coercivity of 235 Oe. The two-step nature of the hysteresis loop can be attributed to presence of multi-domains in the relatively thick flake, as previously reported[30]. We would like to note that the MOKE plots correspond to a flake exposed directly to air for more than a week, indicative of the lack of material degradation (barring surface oxidation which cannot directly be concluded from MOKE) and hence, the air-stability of FGaT nanosheets.

## Deterministic switching of vdW magnet
To demonstrate current induced switching of magnetism in FGaT, we have fabricated bilayer devices of exfoliated FGaT flakes (bottom) and 6 nm sputtered Pt (top). The bilayer stack is patterned into a Hall bar

device and anomalous Hall resistance measurements are used to track the magnetization state of FGaT in the switching experiments. Figure 2e shows the optical image of one such device, D1, with 57.9 nm FGaT and 6 nm Pt patterned into a 5 μm wide Hall bar. The device is subjected to a current waveform as shown in Fig. 2f, with 1 ms write pulses up to a maximum of 6 mA, at 1 s intervals, with $R_{xy}$ being recorded after each current pulse. Figure 2g shows the Hall resistance across this device for the cyclic current sweeps. The measurements are performed at 300 K in the presence of a 100 Oe in-plane magnetic field applied parallel to current direction. The device transitions from a low resistance to a high resistance state and vice versa, for a current pulse of ± 5.4 mA, indicating 180° switching of the OOP magnetization, with appreciable repeatability across four consecutive measurement cycles. This corresponds to a threshold switching current density of 1.69 × $10^6$ A cm$^{-2}$, which compares well with previous reports of ferromagnet/ Pt systems (Supplementary Information Section 3). The small asymmetry in the positive and negative current switching cycles can be attributed to an OOP component of external magnetic field acting on the device due to slight misalignment between the field direction and the true sample plane. We have observed similar current-induced switching behavior in three more of our FGaT/Pt devices (viz. D2, D3, D4) as documented in Section 2 of the Supplementary Information.

As depicted in Fig. 3a, a charge current $I_c$ injected in the plane of the device results in the creation of a transverse spin-current in the Pt layer due to spin-Hall effect. The vertically flowing component of the spin current results in an in-plane oriented spin accumulation at the FGaT/Pt interface. The in-plane damping-like torque applied by the moment of these spins drives the magnetization in-plane while the current pulse is active. The torque from the accompanying external in-plane field, parallel to current direction, favors relaxation of the

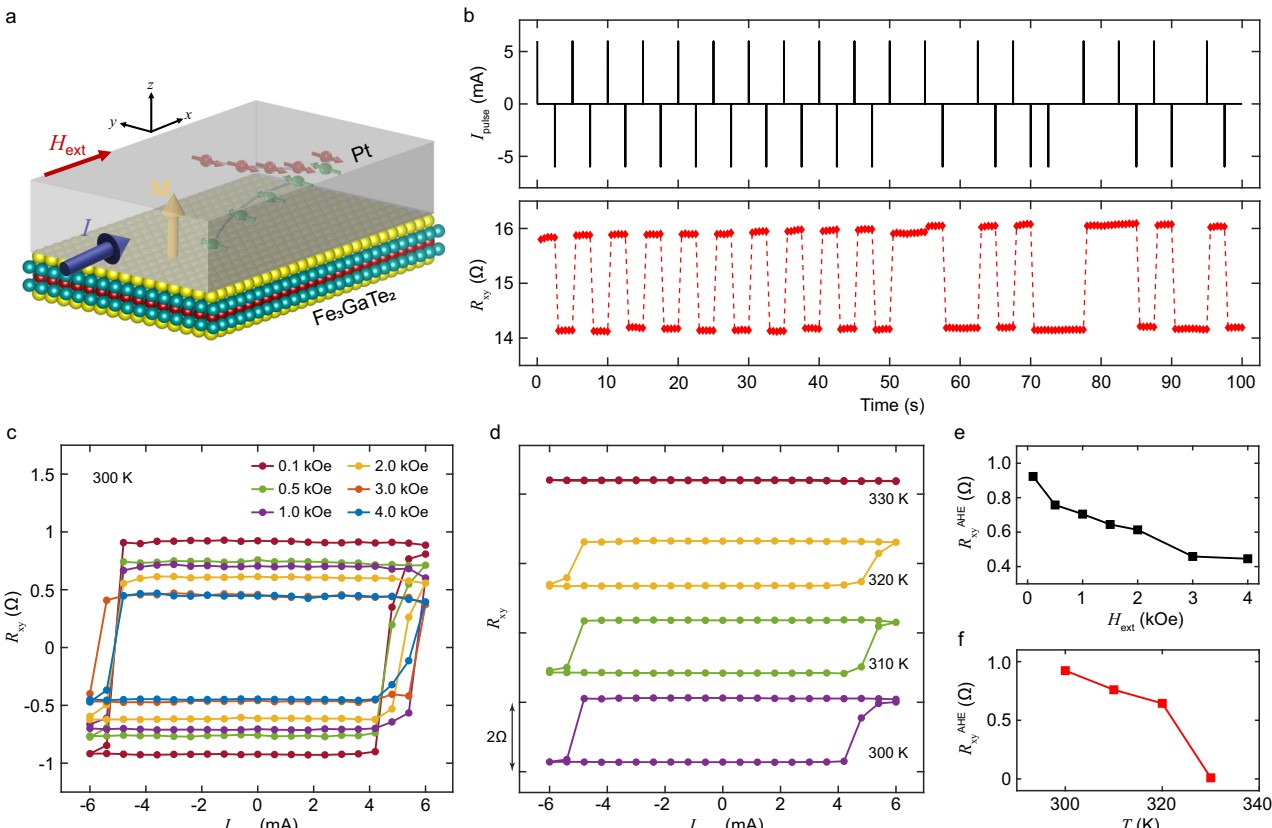

**Fig. 3 | Robust current-driven switching and its field and temperature dependence. a** Schematic diagram showing generation of spin-current in Pt layer in response to a planar current injection ($I$) due to spin-Hall effect. Vertical component of the in-plane polarized spin-current is incident at the Pt/FGaT interface, resulting in spin accumulation and spin-orbit torque on FGaT magnetization **M**. **b** Deterministic, non-volatile switching of FGaT magnetization by a train of current pulses, 1 ms wide and ± 6 mA in magnitude. Modulation of current-induced $R_{xy}$ hysteresis curves upon variation of **c** externally applied in-plane magnetic field magnitude and **d** temperature of the system. Data offset along y-axis for clarity. **e, f** Variation of anomalous Hall resistance against in-plane field **e** and system temperature **f**.

magnetization to one of the two OOP directions (opposite for positive and negative current) after the current pulse is removed[31], resulting in deterministic, non-volatile switching of magnetization in the PMA material. Figure 3b shows the robust control of magnetization state in device D1 at room temperature using a train of random current pulses, 1 ms wide and ± 6 mA in magnitude. We further studied the dependence of magnetization switching curves for varying in-plane field and increasing temperature. In Fig. 3c, we observe that increasing the magnitude of applied in-plane magnetic field results in vertical shrinking of the switching curves. Increasing in-plane magnetic field drives the steady state magnetization of FGaT further away from the $c$-axis, decreasing the OOP magnetization component. Thus, we can expect a reduction in the anomalous Hall resistance of the device ($\propto M_z$) on increasing the in-plane magnetic field, in agreement with our observations (Fig. 3e). Similar trend is also observed in D2 (Supplementary Fig. 3). We also report the response of device D2 to current pulsing, for field variation in other principal axes, i.e., out-of-plane and in-plane (orthogonal to current) in Supplementary Fig. 4, where deterministic switching is not observed. Similarly, we have found that the AHE splitting of the switching curves decreases on increasing the temperature above 300 K (Fig. 3d). We continue to see a sharp transition in resistance up to 320 K, but at 330 K, there is no clear change in $R_{xy}$ across the current cycle. Once again, this agrees well with the expectation that the magnitude of magnetization decreases with increasing temperature and goes to zero beyond the Curie temperature ($\approx$ 328 K). In fact, the trend of $R_{xy}^{AHE}$ observed for these current pulse sweeps (Fig. 3f) matches well with the $R_{xy}^{AHE}$ vs $T$ observed in our FGaT-only devices (Fig. 2b). Similar measurements performed on

device D2, over a boarder temperature range and with finer sampling close to the switching current shows a gradual decrease in threshold switching current with increasing temperature, as is expected due to decrease in magneto-crystalline anisotropy and coercivity at higher temperatures (Supplementary Fig. 3).

## Estimation of Spin-Orbit Torque efficiency

To quantify the spin-orbit torque in our FGaT/Pt bilayer system, we have performed second harmonic Hall (SHH) voltage measurements[32,33]. As depicted in Fig. 4a, the SHH voltage ($V_{xy}^{2\omega}$) is measured along the $y$-axis for an ac current excitation along the $x$-axis, in presence of an externally applied in-plane magnetic field, $H_{ext}$, of varying magnitude and azimuthal angle $\phi$ with respect to the $x$-axis. For an arbitrary orientation of magnetization, with polar angle $\theta_m$ and azimuthal angle $\phi_m$, the transverse resistance contains contributions from the anomalous Hall effect and the planar Hall effect,

$$R_{xy}^{\omega} = R_{xy}^{AHE} \cos\theta_m + R_{xy}^{PHE} \sin^2\theta_m \sin(2\phi_m) \tag{1}$$

where, $R_{xy}^{AHE}$ and $R_{xy}^{PHE}$ are the anomalous Hall resistance and planar Hall resistance of the device, respectively. Upon injecting an AC current into the device, the current induced field-like and anti-damping-like torques drive periodic oscillations of magnetization about its equilibrium position, resulting in harmonic oscillation of the transverse resistance $R_{xy}^{\omega}$. Thus, the recorded transverse voltage $V_{xy}(t) = I_{ac}(t) R_{xy}(t)$ contains a $2\omega$ component which can be detected through lock-in measurement. However, the SHH voltage also has contributions from thermal effects which appear in addition to the

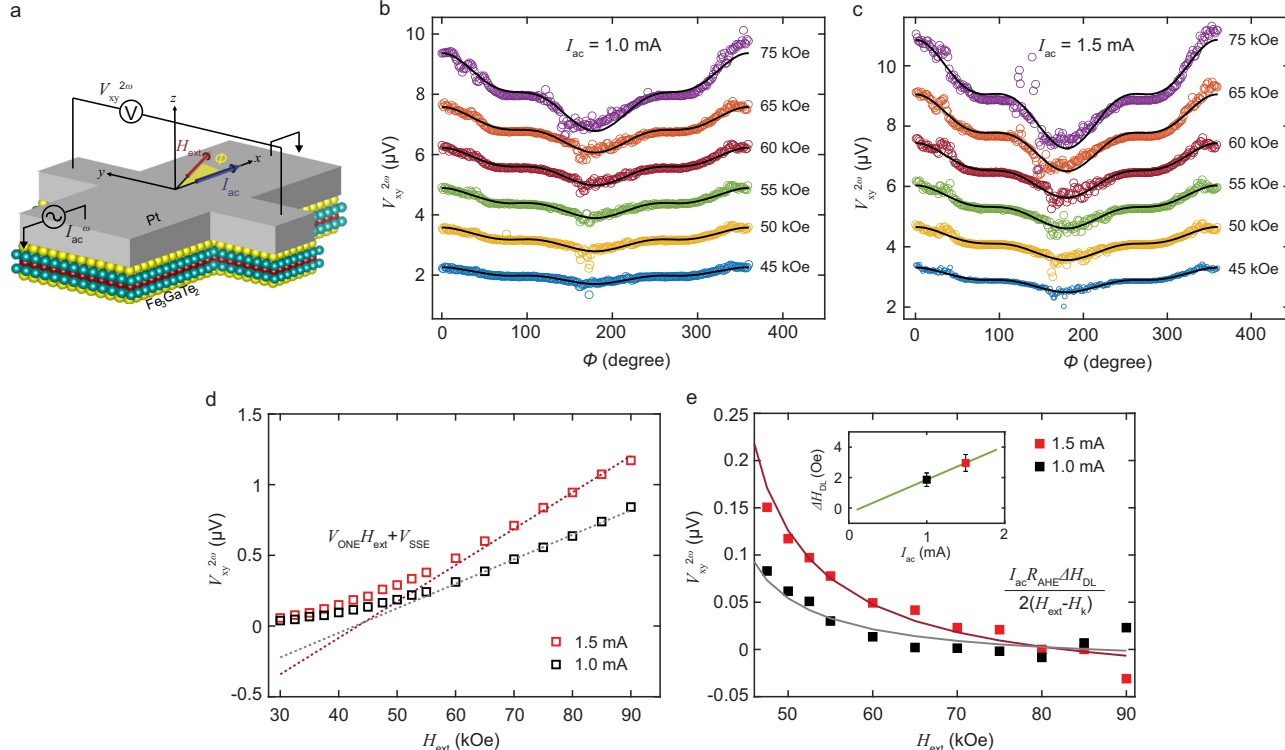

**Fig. 4 | Second harmonic Hall resistance and estimation of SOT efficiency.**
**a** Schematic illustration of the second harmonic Hall (SHH) voltage measurement. External field $H_{ext}$ is applied in the sample ($xy$-) plane at an angle $\phi$ from $x$-axis. Current is applied along $x$-axis and SHH voltage ($V_{xy}^{2\omega}$) is measured along the $y$-axis. **b, c** $V_{xy}^{2\omega}$ measured for in-plane magnetic field rotation for (b) $I_{ac} = 1$ mA and (c) $I_{ac} = 1.5$ mA. Solid black lines fit to Eq. (2). Data offset in y-axis for clarity. **d** Hollow squares represent the amplitude of $\cos\phi$ components of $V_{xy}^{2\omega}$ in Eq. (3). Dotted lines are fits for the linear, thermal contribution to $V_{xy}^{2\omega}$ from ordinary Nernst effect and spin Seebeck effect. **e** Anti-damping-like field contribution to $V_{xy}^{2\omega}$ (solid squares) and their theoretical fit, with $H_k = 38$ kOe. Inset: $\Delta H_{DL}$ extracted for the two current level, and their fitting line, with near zero y-intercept. Error bars represent a 95% confidence interval. Measurements taken at 300 K.

SOT-induced components and can lead to overestimation of SOT efficiency unless systematically eliminated from $V_{xy}^{2\omega}$[34]. Joule heating in the device, proportional to $I_{ac}^2$ (and hence containing $2\omega$ component), creates a vertical thermal gradient. The voltage measured along the $y$-axis, is thus proportional to the component of $H_{ext}$ along $x$-axis through the ordinary Nernst effect (ONE)[35] and to $M_x$ through the anomalous Nernst effect (ANE) and spin-Seebeck effect (SSE). As a result, the SHH voltage takes the form[35],

$$V_{xy}^{2\omega} = P\cos\phi + Q\cos(2\phi)\cos\phi \qquad (2)$$

where,

$$P = \frac{I_{ac}R_{xy}^{AHE}}{2}\frac{\Delta H_{DL}}{H_{ext} - H_k} + V^{ONE}H_{ext} + V^{SSE} \qquad (3)$$

$$Q = I_{ac}R_{xy}^{PHE}\frac{\Delta H_{FL} + H_{Oe}}{H_{ext}} \qquad (4)$$

Here, $\Delta H_{DL}, \Delta H_{FL}$ and $H_{Oe}$ are the effective fields corresponding to current induced anti-damping-like (ADL) torque, field-like torque and Oersted field, respectively. $H_k = 38$ kOe is the effective anisotropy field, $V^{ONE}$ is the SHH voltage per unit applied field and $V^{SSE}$ is the field-independent SSE and ANE contribution.

The SHH voltage corresponding to AC excitation of amplitude 1 mA and 1.5 mA in device D1 is plotted in Figs. 4b and 4c, respectively. The solid black lines correspond to least squared error fit of the recorded voltage to Eq. (2). The $\cos\phi$ components of the voltages,

corresponding to combined ADL torque, ONE and SSE/ANE contribution, is plotted against $H_{ext}$ in Fig. 4d. As can be noted from Eq. (3), the ADL component dwindles upon increasing $H_{ext}$ far over $H_k$ and we expect to see a linear scaling of $V_{xy}^{2\omega}$ at high fields due to the dominant ONE and SSE/ANE. Thus, a linear fit to $V_{xy}^{2\omega}$ at high fields (dotted lines in Fig. 4d) can be used to eliminate thermal contributions from the SHH voltage. Figure 4e shows the SHH voltage corresponding solely to ADL torque, obtained upon subtraction of thermal contributions from $V_{xy}^{2\omega}$ in Fig. 4d. We estimate an anti-damping-like field per unit current density of $\frac{\Delta H_{DL}}{J_c} = 1.34 \times 10^{-10}$ Oe A$^{-1}$ m$^2$, where $J_c$ is the current density in Pt only, calculated based on the parallel resistor model (Supplementary Information Section 1). The anti-damping-like spin torque efficiency can then be calculated as,

$$\xi_{DL} = \left(\frac{2e}{\hbar}\right)M_s t_{FM}\frac{\mu_0\Delta H_{DL}}{J_c}. \qquad (5)$$

Using the bulk saturation magnetization of our FGaT crystals, $M_s = 5.8$ emu g$^{-1}$ (or $3.95 \times 10^4$ A m$^{-1}$), we obtain $\xi_{DL} = 0.093$ for our FGaT/Pt device. Similar measurements in device D2 resulted in $\xi_{DL} = 0.098$ as presented in Supplementary Fig. 5. These values are in good agreement with that expected from a SOT system using Pt as the spin-Hall material with a metallic ferromagnet[36]. We also provide a comparison of threshold switching current density, ADL torque per unit current and SOT efficiency from various previous reports of deterministic magnetization switching in vdW magnetic materials in Supplementary Information Section 3.

## Discussion

We have demonstrated current-induced, deterministic switching (type-z) of out-of-plane magnetization in a $Fe_3GaTe_2$/Pt bilayer spin-orbit torque system up to 320 K, with a low switching current density of $1.69 \times 10^6 \, A \, cm^{-2}$ at room temperature. Furthermore, using second harmonic Hall voltage measurements, we have quantitively estimated the effective anti-damping-like field and spin-torque efficiency of the FGaT/Pt system, and found it to be in good agreement with previous reports of Pt-based SOT systems. Our FGaT/Pt devices not only works beyond room temperature but also achieves switching at one of the lowest current densities (barring the CGT/Pt system[37], where the FM is an insulator and hence not suitable for magnetic tunnel junctions) providing an energy efficient alternative for spintronic devices. While the ADL SOT efficiency for our FGaT/Pt devices ($\approx 0.1$) is close to the commonly observed bulk ferromagnet/Pt systems, the observed switching current density is almost an order of magnitude smaller. This may be attributed to the possible existence of a current-based control of magnetic anisotropy in FGaT, as has been observed in its close analogue, $Fe_3GeTe_2$[38–40]. The existence and extent of this effect in FGaT, however, remains to be explored. During the preparation of this manuscript, we came across an archived report by Li et al. on a similar system of FGaT/Pt[41]. We would like to note that the threshold switching current density we report is almost an order of magnitude smaller than that of Li et al. Furthermore, while their reported $\xi_{DL} = 0.22$ is larger, it may be attributed to oversight of thermal contributions to the SHH voltage even upon operating at current densities larger than ours, as has been argued previously[37] to result in overestimation of SOT effects[18]. By fully accounting for these contributions, we assume that our SOT efficiency value, although smaller, might be a more representative estimate for the FGaT/Pt bilayer system. This work marks an important step in the adoption of vdW magnetic materials for building spintronic devices for non-volatile, energy-efficient memories and computing devices.

## Methods

### Bulk crystal growth and characterization

Single crystals of $Fe_3GaTe_2$ were synthesized through a Te self-flux method as described by Zhang et al.[23]. A mixture of Fe powder (Beantown Chemical, 99.9%), Ga ingot (Alfa Aesar, 99.99999%), and Te pieces (Sigma Aldrich, 99.999%) were weighed in a molar ratio of 1:2:2 in a nitrogen-filled glovebox (with $H_2O$ and $O_2$ levels less than 0.1 ppm) and placed in an alumina Canfield crucible. The mixture-filled crucible was flame-sealed in an evacuated quartz tube with quartz wool and subsequently heated to 1000 °C from room temperature within an hour. The mixture dwelled for 24 h and subsequently cooled to 880 °C within an hour followed by a slow cooling to 780 °C at a rate of 1 °C per hour. Centrifugation was performed to remove the excess flux and afterwards the products were heat-treated to alleviate the concentration of tellurium defects. The resulting products contain a mixture of products with a silver luster among which contain $Fe_3GaTe_2$ crystals which are millimeter sized.

Powder X-ray diffraction (PXRD) data were measured on bulk samples using an X'Pert Pro diffractometer (PANalytical) in Bragg-Brentano geometry operating with a curved Ge(111) monochromator and Cu $K_{\alpha1}$ radiation with a wavelength of 0.154 nm. Scanning electron microscope (SEM) images on both bulk crystals and exfoliated flakes on $SiO_2$ substrates are measured using a Zeiss Merlin high-resolution SEM system with images acquired at an acceleration voltage of 20 kV, a current of 1000 pA, and a working distance of 8.5 mm. SEM-energy dispersive X-ray spectroscopy (EDS) elemental maps were taken using the EDAX APEX software.

Electrical transport measurements were performed on the bulk crystals in a Physical Property Measurement System (PPMS Dynacool, Quantum Design) in a five-probe geometry with contacts made of silver epoxy H20E and platinum wires. Direct current magnetization of

the sample was acquired using the Vibrating Sample Magnetometer (VSM) option on crystals placed in the proper orientation using Kapton tape on a quartz holder (OOP) or using a Brass holder (IP). Temperature-dependent field-cooled and zero-field-cooled magnetization measurements were performed with an applied magnetic field of 1000 Oe. Specific heat capacity was measured on a 4.4 mg crystal using the Heat Capacity (HC) option with Apiezon H vacuum grease applied on the platform.

### Device fabrication

$Fe_3GaTe_2$ flakes were exfoliated on $Si/SiO_2$ dies in a nitrogen-filled glovebox. For FGaT/Pt devices, the dies after exfoliation were sealed in the glovebox and only briefly exposed to air while transferring to the load-lock chamber of the sputterer. An RF plasma cleaning step was performed, under 4 mTorr Ar pressure and 40 W for 15 s, to remove any nascent oxide from FGaT surface. Subsequently, 6 nm Pt was deposited at a rate of $1.8 \, A \, s^{-1}$. 6-terminal Hall bars were patterned into the FGaT/Pt stack through e-beam lithography using negative resist ma-N2403, followed by Ar ion-milling (300 keV, 300 s, 70° inclination). Finally, lift-off through photolithography was used to pattern contact traces and pads for the etched Hall bars. Ti/Au (5 nm/70 nm) was e-beam deposited to create these metallic contacts. FGaT-only devices for magneto-transport characterization were fabricated using e-beam lithography with PMMA resist and Ti/Au (5 nm/35 nm) contacts, ensuring under a minute of direct exposure of FGaT flakes to ambient air. These devices were additionally encapsulated with thick hBN flakes in a glovebox using PDMS-based dry transfer.

### Transport measurements

All transport measurements were performed in a 9 T PPMS DynaCool system. Anomalous Hall effect measurements on the FGaT-only devices were performed using the Electrical Transport Option of the PPMS Dynacool, with a drive current of 100 µA. Current-induced switching experiments were performed by interfacing a Keithley 6221 current source and 2182 A nanovoltmeter with the PPMS Dynacool. Current input sequence consisted of a 1 ms write-pulse followed by 999 ms of read pulses ($\pm 200$ µA). For second harmonic Hall measurements, AC current was supplied by the Keithley 6221, at a frequency 1711.97 Hz. Two lock-in amplifiers, Stanford Research Systems (SRS) SR860 were used to simultaneously measure $f$ and $2f$ transverse voltage components.

## Data availability

Source data are available in the Dryad repository[42], https://doi.org/10.5061/dryad.1rn8pk11k.

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

## Acknowledgements

This work was carried out in part through the use of MIT.Nano's facilities. This work was performed in part at the Harvard University Center for Nanoscale Systems (CNS); a member of the National Nanotechnology Coordinated Infrastructure Network (NNCI).

## Author contributions

S.N.K. and T.N. contributed equally to this work. S.N.K. and D.S. designed the project. T.N., A.B. and M.L. grew the bulk FGaT crystals. T.N., M.L., and C.A.C. characterized the bulk FGaT crystals. S.N.K. fabricated the devices and performed the experiments with assistance from D.S., T.N., M.L., and D.C.B. S.N.K., T.N., and D.S. wrote the manuscript with contributions from all authors.

## Competing interests

The authors declare no competing interests.
