## [Peer Review File · Nature Communications]

Reviewers' Comments:

Reviewer #1:

Remarks to the Author:

Comments:

Shivam N. Kajale et al. reported on the "Current-induced deterministic switching of van der Waals ferromagnet at room temperature" using a Pt/Fe₃GaTe₂ heterostructure device. They demonstrate the current-controlled switching up to 320 K, with low current density of 1.9×10^6 A/cm². The SOT efficiency is estimated to be ~ 0.093 using second harmonic Hall measurement. In general, this work is interesting and I will be happy to see it being accepted. However before recommending its publication, I have several serious comments and questions that need to be addressed:

1. My first concern relates to the novelty of this work, given the abundance of Pt/magnet switching cases in the literature. Room-temperature switching can generally be achieved when combining a room-temperature magnet with Pt. Merely using Fe₃GaTe₂ as a room temperature magnet may not be novel enough to selling the room-temperature spin torque switching.
2. Additionally, the absence of magnetic field-free switching in this work diminishes the importance. The authors should identify unique aspects of their switching compared to previous studies to clarify the novelty issue and strengthen the significance of their work.
3. I noticed at least two other papers on arXiv about Pt/Fe₃GaTe₂ switching, particularly <https://arxiv.org/abs/2304.10718> and <https://arxiv.org/abs/2307.01329>. The latter paper seems to delve further into the spin-torque switching of Pt/Fe₃GaTe₂ devices. Anyway, I believe the authors may need to think hard on the novelty and importance of their work, and adding a comparison discussion could better illustrate the differences of this work.
4. Introducing spin torque into vdW magnets starts from the material Fe₃GeTe₂ with two different types in literature: one is spin-orbit torque in magnet/heavy-metal Fe₃GeTe₂/Pt [Sci. Adv. 5, eaaw8904 (2019); Nano Lett. 19, 4400-4405 (2019)], another is intrinsic spin-orbit torque in single Fe₃GeTe₂ itself [Phys. Rev. Lett. 122, 217203 (2019); Adv. Mater. 33, 2004110 (2021); Adv. Funct. Mater. 31, 2105992 (2021)]. Fe₃GaTe₂ basically shares the same crystalline structure and symmetries as Fe₃GeTe₂, so I am curious how the authors see these two types of spin torques in Fe₃GaTe₂? Do the author perform certain measurements in single Fe₃GaTe₂ itself in addition to this more common Fe₃GaTe₂/Pt case, and get any idea about how much proportion of spin-orbit torque comes from Fe₃GaTe₂ itself while how much results from the Fe₃GaTe₂/Pt interface?

In addition, I have some other concrete concerns on different parts:

5. The term "deterministic" is typically used when the current alone can determine the switching without the need for magnetic fields, which is not the case in this work. It would be better to remove this terminology from the manuscript.
6. Magnetic vdW materials and its spin-orbit torque switching for spintronics have been much developed these few years. But the introduction part didn't include sufficient literature review on this topic, with only one short paragraph. I suggest the authors do more solid literature review and background introduction. Anyway, it is just personal taste and left to the authors' discretion.
7. A thickness of 57.9 nm may not be sufficiently impressive for a magnetic device, given the abundance of vdW magnet/Pt bilayers with much thinner dimensions reported in the literature. The authors should demonstrate similar switching using much thinner Fe₃GaTe₂ (at least below 20 nm), as done in other two arXiv papers on the Fe₃GaTe₂/Pt system.
8. The manuscript lacks discussion of the switching ratio in Fig. 2 and Fig. 3, where the ratio should be calculated as switched R_{xy} by current / full saturated R_{xy} by Hz. In addition, in Fig. 2G, the switched R_{xy} (1.75 Ohm/2) appears larger than the saturated R_{xy} (~ 0.8 Ohm/2 in Fig. 2A), which is abnormal as the switched R_{xy} can never be larger than the maximum saturated R_{xy}. Do the authors know what happen in this situation?
9. In Fig. 3D, it appears that the switching current is smaller for 300 K than 320 K. Typically, a magnetic device requires a smaller switching current at higher temperatures due to the lower magnetic anisotropy at elevated temperatures. I recommend the authors to perform switching measurements over a broader temperature range and address this issue.
10. In my understanding, the V_{1w}-V_{2w} method in arXiv 2304.10718, (2023) is generally considered more common and precise than the V_{2w} under rotation method in Fig. 4. The former method involves the simplest fitting rather than the complicated formula (2). It would be better if the authors could also perform the V_{1w}-V_{2w} measurements on their samples. Just a minor comment, it is up to the authors' decision.

Reviewer #2:

Remarks to the Author:

The authors have found a system of 2 dimensional van der Waals materials that have ferromagnetism at room temperature and demonstrated deterministic switching with platinum adjacent as a spin Hall source. Finding a system working at room temperatures is interesting for the research community and for future applications. I would like to address some questions that were not clear and if the authors clarify these points I think this manuscript can be published in Nature Communications.

1) In the growth and characterization section, the authors showed some specific heat graphs for FGaT samples however it was not clear what the authors wanted to show with the specific heat data. Could the authors explain about the specific heat graph?

2) The authors have shown that the FGaT samples have clear perpendicular magnetic anisotropy (PMA) in this system. Normally, in PMA systems the method to measure the SOT is to tilt the magnetization in the in plane direction and measure the 1st and 2nd harmonic measurements (Nat. Nanotech. 8, 587; Nat. Mater. 12, 240). For these measurements there are some techniques to make sure that the systems do not form multi domains which is a crucial assumption in the analysis. The method in which the authors used to measure the second harmonics is for in-plane samples. Firstly, it is not clear for me why the authors have employed this method. Secondly, when the applied magnetic field increases it gets harder to measure the field like torques. Although the anti-damping torques are the main contribution to switching the field like torques can not be neglected. Have the authors tried these other methods in measuring the spin orbit torques?

Reviewer #3:

Remarks to the Author:

This work presents experimental evidence of current-induced switching in a van der Waals ferromagnet at room temperature. The van der Waals ferromagnet system in this demonstration is a $\text{Fe}_3\text{GaTe}_2/\text{Pt}$ bilayer system. My assessment of the manuscript is as follows:

- The reported results complement those in the existing literature and will be of interest to the readership of Nature Communications. The technical quality of the work is also of the standard expected of the journal. Hence, the article warrants publication. However, it should be suggested that the authors revise their manuscript in response to the remaining points in this reviewer report.

- The measurements of the switching in magnetization were performed at room temperature, which demonstrates the potential of the material system for devices switched by spin-orbit torque. However, it is notable that the device is not capped with an oxide layer like MgO. The AHE signal will be very weak and is unsuitable for real-world applications. Perhaps the authors can comment on whether their material system can be incorporated in a MTJ-like structure and what are the main challenges towards that. I would think achieving a good quality interface with the oxide layer can be detrimental to the SOT efficiency.

- The authors should also clearly state which of the SOT switching schemes is observed and/or characterized: Type-X, Type-Y or Type-Z? More results like those in Fig. 3 but with the external field varied in the three different principal axial directions should be included (perhaps in the supplementary so that the main manuscript is not too long).

- As for the estimation of the spin-torque efficiency of the anti-damping torque, the authors can perhaps refer to the works done by C. L. Chien at John Hopkins University. One of the important findings is that the estimated spin-torque efficiency can be strongly dependent on the measurement and estimation methods used. However, for a fixed method, measurements on

different material systems give a fairly good estimate of the relative differences in the spin-torque efficiency factor. This should help to better clarify how accurate the measured spin-torque efficiency is.

REVIEWER COMMENTS

The authors would like to thank the reviewers for their suggestions which helped us to revise and improve the manuscript. The point-by-point response is given below in blue and all edits in the manuscript and supplementary information have also been marked in yellow.

Reviewer #1 (Remarks to the Author):

Comments:

Shivam N. Kajale et al. reported on the "Current-induced deterministic switching of van der Waals ferromagnet at room temperature" using a Pt/Fe₃GaTe₂ heterostructure device. They demonstrate the current-controlled switching up to 320 K, with low current density of 1.9×10^6 A/cm². The SOT efficiency is estimated to be ~ 0.093 using second harmonic Hall measurement. In general, this work is interesting and I will be happy to see it being accepted. However before recommending its publication, I have several serious comments and questions that need to be addressed:

1. My first concern relates to the novelty of this work, given the abundance of Pt/magnet switching cases in the literature. Room-temperature switching can generally be achieved when combining a room-temperature magnet with Pt. Merely using Fe₃GaTe₂ as a room temperature magnet may not be novel enough to selling the room-temperature spin torque switching.

Response: As the Reviewer has pointed out, spin-orbit torque in Pt/bulk ferromagnet (FM) systems has been studied rigorously over the last decade. However, achieving this in room temperature van der Waals (vdW) FM systems is a significant milestone. To emphasize this point, it is important to first discuss the importance of vdW ferromagnets in the current spintronics landscape and thereafter, how the results of this work may catalyze the harnessing of their technological potential.

vdW FMs provide the path to realizing two-dimensional (2D) spintronic devices which are perhaps the most anticipated step in spintronics since the discovery of spin-orbit torque switching. Use of 2D FMs in spintronic devices can overcome the daunting hurdles in the scaling of bulk FM based spintronic technologies, like degradation of the tunnel barrier in magnetic tunnel junctions (MTJs) through interlayer diffusion and the interface roughness becoming comparable to MTJ layer thicknesses. vdW FMs have also broadened the material space for building spintronic technologies, by providing metallic, perpendicular magnetic anisotropy (PMA) alternatives to the very few bulk options like the CoFeB/MgO system. Furthermore, as several advantages of emerging paradigms of computing like stochastic, neuromorphic computing are unfolding, there is a growing interest in developing ultra-miniaturized, stochastically switching spintronic devices.

2D magnetic materials provide an invaluable balance of scalability and stability needed for achieving this.

Considering the potential transformative effects of vdW ferromagnets in spintronics and hence computing technologies, a solution for electrically controlling them at room temperature is a significant milestone. The first discovery of magnetism in 2D materials came as late as 2017^{1,2}, with ferromagnetism sustained only under liquid nitrogen temperatures. While the discovery of Fe₃GeTe₂ followed by its close analogues³⁻⁵ were pushing the Curie temperature closer to room temperature, they could not provide a solution for room temperature electrical control of ferromagnetism even for the 3-5 years period after their discovery. Through our work, we provide a solution to this crucial hurdle using the metallic vdW FM with PMA, Fe₃GaTe₂ (FGaT), which sustains its ferromagnetism well above room temperature and was reported only as of late 2022⁶. The fact that we can achieve this technologically significant milestone simply using a superior FM does not make it trivial but a reliable and more accessible solution for technological developments, as compared to employing complex tricks like ionic gating, straining, doping, etc. which make translation to commercial products significantly more difficult.

Thus, in view of the importance of vdW magnetic materials in spintronics today, and the effect our results can have on their translation to real world products, we sincerely believe that this work deserves reaching the wide and important audience that a platform like Nature Communications provides.

2. Additionally, the absence of magnetic field-free switching in this work diminishes the importance. The authors should identify unique aspects of their switching compared to previous studies to clarify the novelty issue and strengthen the significance of their work.

Response: We agree with the Reviewer that a field-free switching scheme for FMs is more lucrative as compared to a field-assisted one, and we see that as the next important milestone in the field of vdW spintronics. Yet, this does not in any way diminish the significance of our current results. As discussed in the previous response, this is the first, rigorous demonstration of not just current control, but electrical control (among either current or voltage) of room temperature vdW magnetism. The inability for room temperature operation has been the biggest criticism for proponents of 2D spintronic technologies for the last 5-6 years and this result breaks that barrier to worldly significance. It emphatically establishes that the vdW magnet FGaT can in fact be switched with conventional spin-orbit torques, providing a definite direction for further efforts on achieving field-free switching by reducing uncertainty in the choice of the FM in future systems. Furthermore, the use of Pt as the spin-orbit coupling (SOC) layer in this study is valuable, as it allows us to benchmark the performance of FGaT against other FMs, bulk or cryogenic vdW, since Pt is so widely used for achieving SOT switching in most previous FM candidates.

3. I noticed at least two other papers on arXiv about Pt/Fe₃GaTe₂ switching, particularly <https://arxiv.org/abs/2304.10718> and <https://arxiv.org/abs/2307.01329>. The latter paper seems to delve further into the spin-torque switching of Pt/Fe₃GaTe₂ devices. Anyway, I believe the

authors may need to think hard on the novelty and importance of their work, and adding a comparison discussion could better illustrate the differences of this work.

Response: We acknowledge that these two similar studies on FGaT/Pt SOT switching systems have appeared in quick succession on arXiv.

During the preparation of our manuscript, we had come across the archived report by Li et al. (<https://arxiv.org/abs/2304.10718>) that the Reviewer has mentioned. Our work was done independently, and we have addressed Li et al.'s work in our Conclusion section. The threshold current density for switching that Li et al. have reported ($1.3 \times 10^7 \text{ A cm}^{-2}$) is almost an order of magnitude higher as compared to our device (D1) in the main text ($1.69 \times 10^6 \text{ A cm}^{-2}$), which is also the lowest among all reported metallic vdW FM SOT switching systems (see table in Supplementary Section 3). Furthermore, we have observed and reported similar switching behavior in three more devices (including two newly fabricated and measured during this revision) in the Supplementary Information, compared to Li et al. who have reported the data for just the one device (they mention a second device in the main text, but its data is not disclosed in the arXiv manuscript). Thus, the low current density and reproducibility in our report makes it a superior and more compelling advocate for the FGaT/Pt system.

A difference in rigor of the study is also evident when comparing our manuscript with Li et al. Particularly, we clearly show the response of our device to a train of current pulses creating reliable switching, which is absent in their report. This is important, as quite often the devices degrade after consecutive pulsing, reducing their reliability and showing their response to continuous current pulses is the only way of demonstrating that the devices can produce robust, non-volatile switching. Additionally, we have performed temperature-dependent switching studies and found that switching can be achieved up to 320 K. Li et al. only report switching at 300 K.

While Li et al. have quantitatively characterized the SOT efficiency of the FGaT/Pt system, their method fails to account for the thermal contributions which are well known by now to riddle the second harmonic Hall voltage measurements, as it is a limitation of their method. In fact, they clearly state that they have neglected thermal effects from Joule heating in the SOT efficiency calculation, even though their current densities are higher than what we are operating with. Resultantly, they end up with an anti-damping like SOT efficiency of $\xi_{DL} = 0.22$, which is significantly higher than what Pt based SOT systems can achieve ($\xi_{DL} \approx 0.1$) as they have acknowledged too. On the other hand, we employ an updated and refined second harmonic Hall (SHH) measurement technique that is also used in several recent robust studies⁷⁻¹², where a large field is rotated within the plane of the device, and which allows us to very well separate all the thermal contributions from the measured data^{13,14}. As a result, we end up with a lower $\xi_{DL} = 0.093$, but which is much closer to what is expected from Pt, making our estimation of the SOT efficiency quite possibly more reliable and accurate.

The other report by Yun et al. (<https://arxiv.org/abs/2307.01329>) which appeared after us, achieves SOT switching with a threshold current density of $4.8 \times 10^6 \text{ A cm}^{-2}$, which although is fairly lower than Li et al., it is still almost 3 times higher than what we report. Yun et al. also, neither study the switching of their FGaT/Pt system at different temperatures, and only report results at 300 K, nor show the response of their devices to a train of current pulses. Yun et al. do not even report the switching characteristic against variation of in-plane field, which is quite commonly done in such studies. Finally, the authors do not attempt to characterise the SOT efficiency of the FGaT/Pt system. To their credit, Yun et al. do manage to switch their devices without external fields (Fig. 4) by depositing Pt on FGaT with asymmetric edge coverage. However, it is evident that such a strategy for achieving field-free switching only serves the purpose of a demonstration and is not scalable for real-world applications. This is because achieving asymmetric edge coverage requires sputtering the Pt using an oblique target, without rotating the substrate. Such deposition method can never ensure uniformity at wafer scale as every point on the wafer will end up with a different thickness of platinum (which is well-known to influence SOT performance) based on how close it is to the obliquely placed target.

A noticeable difference between our results and the above two reports was perhaps that they demonstrate the switching in $\approx 20 \text{ nm}$ thick FGaT flakes, while we showed it for thicker flakes, and this was very keenly observed by one of our reviewers. However, in this revision, we have included results from thinner devices as will be discussed in detail under Comment #7.

Although we have argued how our results are quantitatively more reliable and our experiments more rigorous, we do see the alternate reports by Li et al. and Yun et al. in a positive light. Their reports further solidify the repeatability of what we are proposing as a technological solution which is very important.

4. Introducing spin torque into vdW magnets starts from the material Fe_3GeTe_2 with two different types in literature: one is spin-orbit torque in magnet/heavy-metal $\text{Fe}_3\text{GeTe}_2/\text{Pt}$ [Sci. Adv. 5, eaaw8904 (2019); Nano Lett. 19, 4400-4405 (2019)], another is intrinsic spin-orbit torque in single Fe_3GeTe_2 itself [Phys. Rev. Lett. 122, 217203 (2019); Adv. Mater. 33, 2004110 (2021); Adv. Funct. Mater. 31, 2105992 (2021)]. Fe_3GaTe_2 basically shares the same crystalline structure and symmetries as Fe_3GeTe_2 , so I am curious how the authors see these two types of spin torques in Fe_3GaTe_2 ? Do the author perform certain measurements in single Fe_3GaTe_2 itself in addition to this more common $\text{Fe}_3\text{GaTe}_2/\text{Pt}$ case, and get any idea about how much proportion of spin-orbit torque comes from Fe_3GaTe_2 itself while how much results from the $\text{Fe}_3\text{GaTe}_2/\text{Pt}$ interface?

Response: The three papers^{15–17} elucidating the intrinsic “spin-orbit torque” had brought forward some interesting physics that the Fe_3GeTe_2 (FGT) system exhibits, and it is quite reasonable to expect that similar effects to some degree must also be present in the closely related material Fe_3GaTe_2 . Yet it is important to understand that the so-called intrinsic “spin-orbit torque” is phenomenologically different, and hence quantitatively incomparable to the spin-orbit torque efficiency in a FM/heavy-metal (or almost all other SOT) systems.

The FGT intrinsic SOT papers show the modulation of magneto crystalline anisotropy of FGT in the presence of significant current densities, which can be used to switch a ferromagnet with lower current. However, there are two important caveats here: first, that the modulation of anisotropy is not the same as spin-orbit torques and second, the switching herein requires switchable (not a constant bias field) external magnetic fields.

Spin-orbit torque, in its traditional conception, refers to an effective, pseudo-field acting on the ferromagnet due to a current stimulus (usually in an adjacent SOC material). This contrasts with the current-induced anisotropy modulation that is reported in the intrinsic SOT papers, because anisotropy modulation is fundamentally different from creating an effective magnetic field. After all, anisotropy creates a symmetric energy landscape (for example, $m \parallel +z$ and $m \parallel -z$ is degenerate for a PMA system), while effective fields create an asymmetric energy landscape ($m \parallel H$ has lower energy than $m \parallel -H$). This nuance is often overlooked as we loosely refer to anisotropies as effective fields when writing free-energy and magneto-dynamic equations. To sum up, the effect discussed in those references is more of a current-control of magnetic anisotropy (CCMA), analogous to voltage-control of magnetic anisotropy (VCMA). Hence, it faces similar challenges as VCMA in independently (without external field control) achieving thermodynamic switching of ferromagnets. This is also evident in the two experimental papers^{15,17} of these three. Although the authors have shown the effectiveness of this CCMA in switching the PMA magnetization with appreciably low current densities, they also need to toggle the external field between positive and negative fields to make the switching possible. Switching from 0 to 1 requires a positive OOP field and 1 to 0 requires a negative OOP field. Thus, we need field control in addition to current control which we argue is effectively a step back, since we want to move away from field control of magnetism toward utilization of electrical-only control. Also, the field needs to be at a sweet spot which can be detrimental for technological applications where device-to-device variability can easily hamper the results.

However, this also means that just as VCMA can assist the switching of PMA magnetization, the CCMA can also play a role in switching FGT and FGaT with lower current densities. Perhaps, this can explain why we are able to achieve switching in our FGaT/Pt system with much lower current density compared to bulk FM/Pt system (typically 10s of MA/cm²) even though the SOT efficiency here is very close to typical bulk FM systems. In our revised manuscript, we have added this discussion as a plausible explanation for the low current densities observed, and we thank the Reviewer for motivating us towards this point.

In addition, I have some other concrete concerns on different parts:

5. The term "deterministic" is typically used when the current alone can determine the switching without the need for magnetic fields, which is not the case in this work. It would be better to remove this terminology from the manuscript.

Response: A switching scheme is categorized as “deterministic” if the final state of magnetization is accurately determined by the initial conditions and the applied stimuli. This contrasts with the case of non-deterministic switching, where the final state cannot be accurately predicted despite controlling the initial state and stimuli, and involves a degree of randomness or stochasticity. Non-deterministic switching can occur when certain conditions for deterministic switching are not met. For example, without certain conditions such as mirror symmetry breaking, chiral symmetry breaking, exchange bias, and interlayer exchange coupling, the switching process may not be deterministic. With regards to the case of conventional SOT switching of a PMA magnet, applying an in-plane magnetic field is what makes the switching “deterministic” by breaking the symmetry between the up and down magnetization states, while in absence of the external field, the final state can be either of the two, based on where the thermal fluctuations swing the magnetization at the end of the current pulse. This point is also schematically explained in Fig. 1 below.

Such categorization of field-assisted switching as “deterministic” is clearly found in the pioneering work of Miron, Gambardella et al.¹⁸ which was one of the first demonstrations of spin-orbit torque switching of a PMA ferromagnet. Making this distinction clear has become increasingly important recently with growing reports of non-thermodynamic switching methods and stochastic switching devices.

It is worth noting that applying an external in-plane magnetic field is perhaps the most common way of ensuring deterministic switching. Hence, it can be argued that stressing the switching scheme as being deterministic becomes more important when the method is field-free. Perhaps, this is what may be creating a perception in contemporary discourse that “deterministic” should mean field-free. Considering that there exists room for such misinterpretation as the Reviewer has kindly pointed out in their comment, we have decided to drop the term “deterministic” from the title. Nevertheless, we respectfully maintain that our usage of the term “deterministic” switching in the manuscript is appropriate and appears in the manuscript when warranted.

Figure 1: (a) A current pulse (J_c) in the FM/heavy metal bilayer exerts an in-plane anti-damping-like torque τ_{DL}^{SOT} on the FM magnetization, orthogonal to the current direction. When the current pulse is sufficiently strong and long, the magnetization is driven in-plane by the end of the current pulse. When the current pulse is terminated, and since there is no external field, the magnetization tends to relax out-of-plane once

again, owing to the FM's perpendicular magnetic anisotropy. However, both up and down magnetization states are equally favorable, and the final state is determined by how the thermal fluctuations swing the magnetization at that instance, resulting in non-deterministic switching. (b) In the presence of a sufficiently strong external field parallel to the current, the field-like torque from the external field, τ_{FL}^{ext} , tends to precess the magnetization either clockwise or counter-clockwise (based on field being parallel or antiparallel to the current) at the instance when the current pulse is terminated. Thus, the magnetization relaxes downwards (in this case) deterministically. Hence, the in-plane field acts as a means to achieving "deterministic" switching of a PMA magnet using conventional SOT from a heavy metal.

6. Magnetic vdW materials and its spin-orbit torque switching for spintronics have been much developed these few years. But the introduction part didn't include sufficient literature review on this topic, with only one short paragraph. I suggest the authors do more solid literature review and background introduction. Anyway, it is just personal taste and left to the authors' discretion.

Response: In accordance with the Reviewer's suggestion, we have expanded our discussion of relevant literature in the Introduction section of our revised manuscript. We thank the Reviewer for their valuable suggestion.

7. A thickness of 57.9 nm may not be sufficiently impressive for a magnetic device, given the abundance of vdW magnet/Pt bilayers with much thinner dimensions reported in the literature. The authors should demonstrate similar switching using much thinner Fe₃GaTe₂ (at least below 20 nm), as done in other two arXiv papers on the Fe₃GaTe₂/Pt system.

Response: We agree with the Reviewer's observation that vdW magnet-based studies often try to report results from under 20-25 nm thickness. So, while we can expect hardly any changes in the physics of the system in going from 50-60 nm to ≈ 20 nm, we have worked in the revision period to align with this trend in the field too. We are happy to report that we could fabricate and observe similar current-induced switching in two new devices where the thickness of the FGaT flake is < 20 nm.

Fig. 2 and Fig. 3 below present the data corresponding to the two new devices, which we refer to as D2 and D3, respectively (D1 is the device in the main text from the original submission). D2 and D3 are fabricated using FGaT flakes which were 17.48 nm and 19.11 nm thick, respectively. Optical images of the flakes before fabrication, the fabricated devices, AFM topology images and AFM height profiles for the two devices are presented in Fig. 2a and Fig. 3a. Both the devices exhibit current-induced switching curves in the presence of an externally applied in-plane field parallel to the current injection direction, as seen in Fig. 2c and Fig. 3c. The threshold switching current density for these devices is found to be 4.23×10^6 A cm⁻² for D2 and 9.54×10^6 A cm⁻² for D3, which is comparable to D1's reported value of 1.69×10^6 A cm⁻². Relevant design and observed parameters used in the calculations are also tabulated in Table 1 for the purposes of this revision. The switching data of these devices have also been included in the Supplementary Information of the revised manuscript.

Figure 2: (a) Morphology of device D2 – optical image of the (i) exfoliated flake and (ii) the corresponding device after fabrication. (iii) AFM topography of the flake, for the dotted yellow box area in (i), and (iv) AFM height profile along the solid yellow line (iii). Scale bars – 5 μm . (b) Anomalous Hall effect measurement of the device at 300 K, with external magnetic field swept out-of-plane ($H \parallel c$) (c) Current-induced switching loop of the device at 300 K with 100 Oe external field applied parallel to the current injection direction ($H \parallel I$). Dashed black lines in (b) and (c) are visual guides for the maximum anomalous Hall resistance achievable in the device at 300 K.

Figure 3: (a) Morphology of device D3 – optical image of the (i) exfoliated flake and (ii) the corresponding device after fabrication. (iii) AFM topology of the flake and (iv) AFM height profile along the solid yellow line (iii). Scale bars – 5 μm . (b) Anomalous Hall effect measurement of the device at 300 K, with external magnetic field swept out-of-plane ($H \parallel c$) (c) Current-induced switching loop of the device at 300 K with 500 Oe external field applied parallel to the current injection direction ($H \parallel I$). Dashed black lines in (b) and (c) are visual guides for the maximum anomalous Hall resistance achievable in the device at 300 K.

Table 1: Parametric details corresponding to the newly included devices D2 and D3.

Device	Hall bar width (μm)	FGaT thickness (nm)	Switching current (mA)	Switching current density (A cm^{-2})	Maximum R_{xy}^{AHE} (mΩ)	Switched R_{xy}^{AHE} (mΩ)	Switching ratio
D2	6.77	17.48	6.72	4.23×10^6	46.35	28.3	0.61
D3	3.55	19.11	8.50	9.54×10^6	38.05	30.0	0.79

8. The manuscript lacks discussion of the switching ratio in Fig. 2 and Fig. 3, where the ratio should be calculated as switched R_{xy} by current / full saturated R_{xy} by Hz. In addition, in Fig. 2G, the

switched R_{xy} (1.75 Ohm/2) appears larger than the saturated R_{xy} (~ 0.8 Ohm/2 in Fig. 2A), which is abnormal as the switched R_{xy} can never be larger than the maximum saturated R_{xy} . Do the authors know what happens in this situation?

Response: We thank the Reviewer for raising this point. We would like to first clarify that the data in Fig. 2a and Fig. 2g are from two different devices. Fig. 2a corresponds to a FGaT-only device, while Fig. 2g corresponds to a FGaT-Pt device (D1). This is reflected in the figure caption as well as the main text. However, we did make a mistake in the schematic inset of Fig. 2a where we have depicted a Pt layer as well, even though it is a FGaT-only device. We believe that this was the source of the confusion, and we sincerely apologize to the Reviewers and the Editor for this lapse. This was an oversight and in no way an attempt to mislead the readers, and we are grateful to the Reviewer for their careful review. We have corrected the schematic in the revised manuscript.

Indeed, as the Reviewer has mentioned, the switched R_{xy} should not be larger than the maximum saturated R_{xy} . The discrepancy described by the Reviewer should not exist in our devices and the two values referred to in the comment originate from different devices. In the initial round of measuring device D1, we had not collected the anomalous Hall effect (AHE) data for the device and upon receiving the Reviewer's comment, we attempted to measure the device again. However, unfortunately, somewhere in the process of swapping the carrier PCB and loading the device in the cryostat again, the device suffered from ESD damage and could not produce any more data. Thus, in all the new devices that we have since measured, we made sure to measure the AHE prior to performing any switching or second harmonic measurements. The AHE plots for the devices D2 and D3 are presented here in Fig. 2b and Fig. 3b, respectively. The switching ratio obtained for the two devices is 0.61 and 0.79, respectively, and the details of the calculation are also tabulated in Table 1. The results are consistent with the essential idea that the switched R_{xy} must be equal to or lower than the maximum saturated R_{xy} .

9. In Fig. 3D, it appears that the switching current is smaller for 300 K than 320 K. Typically, a magnetic device requires a smaller switching current at higher temperatures due to the lower magnetic anisotropy at elevated temperatures. I recommend the authors to perform switching measurements over a broader temperature range and address this issue.

Response: We agree with the Reviewer's perspective and concern described here. It is often seen that the threshold switching current in a SOT device decreases on increasing temperature, owing to a reduction of magnetic anisotropy and coercivity. To resolve this issue, we have performed temperature dependent switching experiments with one of our new devices (D2) over a larger temperature range as suggested by the Reviewer, and with twice the temperature steps above 300 K. Furthermore, we have increased the sampling resolution near the switching current in the new experiments, instead of using uniform sampling across the entire current pulsing range as before, to effectively capture even small trends if present.

Figure 4: (a) Temperature dependent current-induced switching curves for device D2, under 500 Oe external field applied parallel to the current. (b) Representative figure showing the sigmoidal fitting of positive and negative transition edge of the switching loops to extract threshold switching current values. (c) Variation of switched anomalous Hall resistance (R_{xy}^{AHE}) and threshold switching current (I_{sw}) for the device at different temperatures.

Fig. 4a shows the temperature dependent switching data for device D2. The data corresponding to 320 K and 325 K has been scaled for clarity. As is evident, the sampling is denser around the switching current making it possible to capture the fine decrease in switching current with temperature above 300 K. To rigorously estimate the switching temperature for each temperature, we have fitted sigmoidal functions at the positive and negative switching edge of the switching loop, as exemplified for 320 K in Fig. 4b. The average of the transition points for the positive and negative edge is plotted in Fig. 4c in solid blue triangles, as the threshold switching current. Thus, we can now observe that the expected trend of decreasing switching current at higher temperatures, although small, is present. Theoretically explaining how sharp or gradual this trend should be for a FM is a complex task, if even possible, as it has a dependence on the material's coercivity which is known to be difficult to model accurately for a real world nanomagnet. Also, we have limited our measurements up to 10 mA current pulses to avoid damaging the device for further measurements, and hence we have chosen not to go to lower temperatures where higher current would be needed for switching. Yet, the reported

temperature range is adequately and unambiguously showing the expected trend around room temperature. We would like to highlight that the scaling of R_{xy}^{AHE} with temperature matches well with that of device D1 (main text Fig. 3F) where R_{xy}^{AHE} decreases with increasing temperature until no clear switching loop is observed starting 330 K which coincides with the Curie temperature of our FGaT nanosheets. This data (Fig. 4a, 4c) is also included in the Supplementary Information of our revised manuscript. In light of these observations, the apparent non-monotonic variation of switching current for device D1 seen in Fig. 3D (of the manuscript) may be attributed to the sparser spacing between data points thereby creating a larger error margin, incapable of capturing the fine trends in switching current variation.

10. In my understanding, the V1w-V2w method in arXiv 2304.10718, (2023) is generally considered more common and precise than the V2w under rotation method in Fig. 4. The former method involves the simplest fitting rather than the complicated formula (2). It would be better if the authors could also perform the V1w-V2w measurements on their samples. Just a minor comment, it is up to the authors' decision.

Response: The Reviewer has raised an important point regarding our choice of the SOT efficiency (ξ) measurement technique, and we appreciate the opportunity to elaborate on this. Given the wide variety of methods that can be found in literature for measuring SOT efficiencies¹⁹, like spin-torque ferromagnetic resonance (ST-FMR), harmonic Hall voltage response (HHVR), inverse spin-Hall effect (ISHE), hysteresis loop shift and spin Hall magnetoresistance, rationalizing the most appropriate and accurate choice for a particular material system becomes important. Among these, the HHVR technique is well suited and favored for vdW magnet-based SOT systems as is evident from its use in most of the previously reported vdW FM/heavy metal bilayers.

The earliest reports of the HHVR method for estimating SOT efficiency came from the pioneering works of Kim et al.²⁰ and Garello et al. in 2013²¹. The report by Li et al. (arXiv 2304.10718, 2023) closely follows the method proposed by Kim et al. However, as has become clear over the subsequent years, the original papers by Kim et al. and Garello et al. did not consider the various magneto-thermal effects like anomalous Nernst effect (ANE), spin-Seebeck effect (SSE) and the ordinary Nernst effect (ONE) which also generate a second harmonic Hall voltage in the measurement configuration used for SOT efficiency measurement. The presence of in-plane magnetization and field components, coupled with vertical thermal gradients created due to Joule heating and vertical asymmetry in the device environment, matched the conditions needed for observing ANE, SSE and ONE. Consequently, the same group of authors (Avci, Gambardella et al.) published an article in 2014¹³ which advocates a second harmonic Hall response technique where a large external field is rotated in-plane instead of sweeping the field out-of-plane, to be able to eliminate the magneto-thermal components. They clearly show that this method can effectively handle PMA FMs as long as the in-plane field magnitude is large enough to saturate the magnetization in-plane ($H_{ext} > H_{k,eff}$), thus, avoiding multi-domain dynamics. In compliance

with this condition, our measurements are also performed for field magnitudes higher than 38 kOe (found to be the $H_{k,eff}$ of FGaT, see main text Fig. 2c) to ensure mono-domain dynamics of magnetization. Finally, an article in 2019 by Roschewsky, Salahuddin et al. showcased the presence of ordinary Nernst effect too in bilayer SOT systems¹⁴, and extended the model of Garello, Gambardella et al. to contain ANE, SSE and ONE.

Figure 5: Spin-orbit torque efficiency of D2. (a, b, c) The second harmonic Hall voltage, $V_{xy}^{2\omega}$, measured under in-plane magnetic field rotation for (a) $I_{ac} = 1.0$ mA, (b) $I_{ac} = 1.5$ mA and (c) $I_{ac} = 2.0$ mA. Solid black lines fit to equation (2). Data offset in y-axis for clarity. (d) Hollow symbols represent the amplitude of $\cos \phi$ components of $V_{xy}^{2\omega}$ in equation (3). Solid lines are fits for the linear, thermal contribution to $V_{xy}^{2\omega}$ from ordinary Nernst effect and spin Seebeck effect. (e) Anti-damping-like field contribution to $V_{xy}^{2\omega}$ (solid symbols) and their theoretical fit (solid lines), with $H_k = 38$ kOe. Inset: ΔH_{DL} extracted for the three current amplitudes, and their fitting line, with a near zero y-intercept. Error bars represent a 95% confidence interval. The damping-like field is estimated to be 4.66×10^{-10} Oe $A^{-1}m^2$, and the anti-damping-like spin-orbit torque efficiency is $\xi_{DL} = 0.098$. Measurements correspond to 300 K. Refer to main text for fitting equations.

These refinements to the HHVR technique have been well received in the spintronics community and their adoption in recent PMA FM based SOT systems is quite evident in the recent literature. The following are a few recent examples where this technique of HHVR measurements has been used for PMA FM based SOT systems:

1. Gupta, Ralph et al. Nano Lett. 20, 7482–7488, (2020) – This article uses angle-dependent HHVR measurements to estimate ξ_{DL} in the $\text{Cr}_2\text{Ge}_2\text{Te}_6/\text{Pt}$ system. It is a close analogue of our material system, where a vdW ferromagnet with PMA is coupled with the heavy metal Pt¹¹.
2. Alghamdi et al. Nano Lett. 19, 4400–4405 (2019) – Another example where ξ_{DL} for a bilayer system of a vdW PMA ferromagnet (Fe_3GeTe_2) and heavy metal (Pt) is estimated using the angle-dependent HHVR measurement¹².
3. CoFeB/Ta-W/W – Cha et al. Mater. Res. Express 8,106102 (2021) – PMA ferromagnet system CoFeB/MgO with Ta or W/Ta-W underlayers¹⁰.
4. Husain et al. Appl. Phys. Lett. 122, 062403 (2023) – PMA insulating ferrimagnet $\text{Tm}_3\text{Fe}_5\text{O}_{12}$ with heavy metal W⁸.
5. Li et al. Phys. Rev. B 95, 241305(R) (2017) – PMA insulating ferrimagnet $\text{Tm}_3\text{Fe}_5\text{O}_{12}$ with heavy metal Pt⁷.
6. Dutta, Ralph et al. Phys. Rev. B 103, 184416 (2021) – PMA CoFeB with heavy metal Ir⁹.

Thus, considering the theoretical soundness of the angle dependent HHVR technique and its widespread adoption in recent literature, even for a strikingly similar material system, we feel confident that our method is very well suited for the estimation of SOT efficiency in the FGaT/Pt system. In fact, the SOT efficiency that we have estimated for our FGaT/Pt device (0.093) lies close to the value found in most FM/Pt systems, of ≈ 0.1 , compared to the value reported by Li et al. (0.22) which indicates a likely overestimation from ignoring the thermal contributions to SHH voltages. Furthermore, we have repeated these measurements for our newly fabrication device (D2) and found the SOT efficiency in close agreement with the first device ($\xi_{DL} = 0.098$), as a further indication of the reliability of this method. The details of the corresponding measurements are also included in Supplementary Figure 5, and in Fig. 5 above.

Reviewer #2 (Remarks to the Author):

The authors have found a system of 2 dimensional van der Waals materials that have ferromagnetism at room temperature and demonstrated deterministic switching with platinum adjacent as a spin Hall source. Finding a system working at room temperatures is interesting for the research community and for future applications. I would like to address some questions that were not clear and if the authors clarify these points I think this manuscript can be published in Nature Communications.

1) In the growth and characterization section, the authors showed some specific heat graphs for FGaT samples however it was not clear what the authors wanted to show with the specific heat data. Could the authors explain about the specific heat graph?

Response: The plot of the specific heat in Fig. 1g serves as additional thermodynamic experimental evidence of the above room-temperature magnetic transition of Fe_3GaTe_2 in bulk form. The specific heat data shows a slight upturn near ~ 340 K (magnified in the inset). The upturn is relatively small due to the layered nature of the material but nevertheless contrasts the general downward trend for decreasing temperature, with (red) and without (black) applied magnetic field. Similar studies have been performed on Fe_3GeTe_2 (*J. Phys. Soc. Jpn.* **82**, 124711 (2013), *Phys. Rev. B* **93**, 144404 (2016)) to complement electric transport and magnetization measurements on the subject of determining the magnetic transition temperature. To refine these points, we have added some language in the caption to highlight that the purpose of the specific heat measurement was to indicate the above room temperature magnetic transition via another experimental probe. We thank the reviewer on their comment for clarification.

2) The authors have shown that the FGaT samples have clear perpendicular magnetic anisotropy (PMA) in this system. Normally, in PMA systems the method to measure the SOT is to tilt the magnetization in the in plane direction and measure the 1st and 2nd harmonic measurements (*Nat. Nanotech.* **8**, 587; *Nat. Mater.* **12**, 240). For these measurements there are some techniques to make sure that the systems do not form multi domains which is a crucial assumption in the analysis. The method in which the authors used to measure the second harmonics is for in-plane samples. Firstly, it is not clear for me why the authors have employed this method. Secondly, when the applied magnetic field increases it gets harder to measure the field like torques. Although the anti-damping torques are the main contribution to switching the field like torques can not be neglected. Have the authors tried these other methods in measuring the spin orbit torques?

Response: The Reviewer has raised an important point regarding our choice of the SOT efficiency (ξ) measurement technique, and we thank them for giving us a chance to elaborate on this. Given the wide variety of methods that can be found in literature for measuring SOT efficiencies¹⁹, like

spin-torque ferromagnetic resonance (ST-FMR), harmonic Hall voltage response (HHVR), inverse spin-Hall effect (ISHE), hysteresis loop shift and spin Hall magnetoresistance, rationalizing the most appropriate and accurate choice for a particular material system becomes important. Among these, the harmonic Hall voltage measurement technique is well suited for the vdW PMA FM based SOT system like ours. It provides the advantage of resulting in reliable estimation of ξ by allowing separation of false signals contributed by thermal effects, being a widely used method allowing seamless benchmarking against other SOT systems, using the same device structure as is needed for AHE measurements (and by extension, SOT switching), and easy integration with conventional cryostat setups (unlike RF techniques). Thus, we have chosen the harmonic Hall voltage response method to characterize ξ_{DL} for our FGaT/Pt system too, just as most other vdW FM/heavy metal systems reported before (see Supplementary Information Section 3).

The HHVR technique was introduced by the pioneering work of Garello, Gambardella et al.²¹ and Kim, Ohno et al.²⁰ (the same articles referred by the Reviewer), back in 2013. It is based on the principle that the harmonic oscillation of magnetization in response to the SOTs from an ac current (frequency f) excitation, result in a harmonically oscillating planar and anomalous Hall resistance. Thus, the Hall voltage, which is the product of the Hall resistance and the ac current, contains a second harmonic ($2f$) voltage components which can be used to estimate the SOT strength. However, it is worth noting that this technique requires driving the magnetization of a PMA FM towards in-plane orientation using in-plane external fields. The presence of in-plane magnetization and field components, coupled with vertical thermal gradients created due to Joule heating and vertical asymmetry in the device environment, generate second harmonic Hall voltages from anomalous Nernst effect (ANE), spin-Seebeck effect (SSE) and ordinary Nernst effect (ONE). The presence of these effects can severely distort the estimation of ξ , and hence, the effective use of HHVR method requires a careful treatment of all such spurious effects.

While the article by Kim, Ohno et al. did not provide a solution for treating the magneto-thermal effect, Garello, Gambardella et al. had a partial discussion (Supplementary Section 7)²¹ regarding the presence of ANE. However, their method for measuring the ANE separately did not follow the symmetries of a PMA system whose magnetization is tilted towards in-plane direction (they perform ANE measurements for out-of-plane field sweeps on a PMA magnet). As an improvement to this method, the same group of researchers published an article in 2014 (Avci, Gambardella et al.)¹³ which advocates a second harmonic Hall response technique where a large external field is rotated in-plane instead of sweeping the field out-of-plane, to be able to eliminate the magneto-thermal components. They clearly show that this method can effectively handle PMA FMs as long as the in-plane field magnitude is large enough to saturate the magnetization in-plane ($H_{ext} > H_{k,eff}$), thus, avoiding multi-domain dynamics. This is also exemplified in the same article (Fig. 4, 5, 6) for Pt/Co, Ta/Co PMA systems where a field > 162 mT is rotated in-plane to ensure in-plane saturation of Co magnetization. In compliance with this condition, our measurements are also performed for fields higher than 38 kOe (found to be the $H_{k,eff}$ of FGaT, see main text Fig. 2c) to

ensure mono-domain dynamics of magnetization. Finally, an article in 2019 by Roschewsky, Salahuddin et al.¹⁴ showcased the presence of ordinary Nernst effect too in bilayer SOT systems, and extended the model of Garello, Gambardella et al. to contain ANE, SSE and ONE.

These refinements to the HHVR technique have been well received in the spintronics community and their adoption in recent PMA FM based SOT systems is quite evident in the recent literature. Following are a few recent examples where this technique of HHVR measurements has been used for PMA FM based SOT systems:

1. Gupta, Ralph et al. Nano Lett. 20, 7482–7488 (**2020**) – This article uses angle-dependent HHVR measurements to estimate ξ_{DL} in the $\text{Cr}_2\text{Ge}_2\text{Te}_6/\text{Pt}$ system. It is a close analogue of our material system, where a vdW ferromagnet with PMA is coupled with the heavy metal Pt¹¹.
2. Alghamdi et al. Nano Lett. 19, 4400–4405 (**2019**) – Another example where ξ_{DL} for a bilayer system of a vdW PMA ferromagnet (Fe_3GeTe_2) and heavy metal (Pt) is estimated using the angle-dependent HHVR measurement¹².
3. CoFeB/Ta-W/W – Cha et al. Mater. Res. Express 8,106102 (**2021**) – PMA ferromagnet system CoFeB/MgO with Ta or W/Ta-W underlayers¹⁰.
4. Husain et al. Appl. Phys. Lett. 122, 062403 (**2023**) – PMA insulating ferrimagnet $\text{Tm}_3\text{Fe}_5\text{O}_{12}$ with heavy metal W⁸.
5. Li et al. Phys. Rev. B 95, 241305(R) (**2017**) – PMA insulating ferrimagnet $\text{Tm}_3\text{Fe}_5\text{O}_{12}$ with heavy metal Pt⁷.
6. Dutta, Ralph et al. Phys. Rev. B 103, 184416 (**2021**) – PMA CoFeB with heavy metal Ir.⁹

Thus, considering the theoretical soundness of the angle-dependent HHVR technique and its widespread adoption in recent literature, even for a strikingly similar material system, we feel confident that our method is very well suited for the estimation of SOT efficiency in the FGaT/Pt system.

Reviewer #3 (Remarks to the Author):

This work presents experimental evidence of current-induced switching in a van der Waals ferromagnet at room temperature. The van der Waals ferromagnet system in this demonstration is a Fe₃GaTe₂/Pt bilayer system. My assessment of the manuscript is as follows:

- The reported results complement those in the existing literature and will be of interest to the readership of Nature Communications. The technical quality of the work is also of the standard expected of the journal. Hence, the article warrants publication. However, it should be suggested that the authors revise their manuscript in response to the remaining points in this reviewer report.

- The measurements of the switching in magnetization were performed at room temperature, which demonstrates the potential of the material system for devices switched by spin-orbit torque. However, it is notable that the device is not capped with an oxide layer like MgO. The AHE signal will be very weak and is unsuitable for real-world applications. Perhaps the authors can comment on whether their material system can be incorporated in a MTJ-like structure and what are the main challenges towards that. I would think achieving a good quality interface with the oxide layer can be detrimental to the SOT efficiency.

Response: The Reviewer has raised a crucial point which relates to the translation of our reported SOT switching system to mature technologies. We would first like to clarify that our vision for this is the eventual integration of the FGaT-based SOT system in a magnetic tunnel junction (MTJ) stack as the free layer, and not as a stand-alone device where the anomalous Hall resistance encodes the state of the magnet. The latter has disadvantages like a weak signal (as the Reviewer has correctly stated) as well as the need for a 4-terminal device which is detrimental to scalable and compact chip design. Our report uses the anomalous Hall resistance only as a probe for the PMA magnetization state through current-induced switching.

This brings us to their next point regarding the absence of MgO (or a tunnel barrier) layer in our devices and whether adding it can be detrimental to the SOT efficiency. In this regard, it is worth noting that vdW FM based spin-valves (and their subset, MTJs) have most often been designed with a vdW non-magnetic spacer layer, and not a bulk oxide like MgO. MgO, in particular, has gained popularity due to the superior results (large tunneling magnetoresistance (TMR), and strong PMA) it produces in the specific material combination of CoFeB/MgO owing to a special orbital hybridization that occurs at the CoFeB/MgO interface²². Thus, it is highly contentious whether MgO is also the most suitable candidate for a tunnel barrier in vdW FM-based MTJs too. On the contrary, the use of vdW spacer layers allows leveraging the various advantages that vdW materials provide towards device scaling while maintaining superior device performance through pristine layer boundaries and atomically sharp interfaces.

Regarding the possibility of using FGaT in spin-valve structures, we are happy to report here that several experimental articles on this topic have appeared within the last year, even though the material was first reported only in August 2022. We have listed these articles in Table 2. It is encouraging to see that the FGaT-based spin-valves are consistently able to exhibit room temperature TMR (or giant magnetoresistance, GMR) including the impressive 85% TMR at 300 K in FGaT/WSe₂/FGaT MTJs. We have also included this point in the Introduction of our revised manuscript, to provide a better context for our readers. Such rapid and impressive progress in devices design using FGaT, coupled with our demonstration of above room temperature SOT control make us optimistic regarding the technological viability of developing FGaT based spintronic devices.

Table 2: Experimental reports of FGaT based spin-valves (including MTJs).

Material stack	Tunnel Barrier Thickness	Peak GMR/TMR (Temperature)	Author
FGaT/WSe ₂ /FGaT	7 nm WSe ₂	85 % OOP (300 K) 164 % OOP (10 K)	Zhu et al. ²³
FGaT/WS ₂ /FGaT	4.2 nm WS ₂	11 % OOP (300 K) 213 % OOP (10 K)	Jin et al. ²⁴
FGaT/MoS ₂ /FGaT	4.5 nm MoS ₂	0.31 % OOP (300 K) 15.89 % OOP (2.3 K)	Jin et al. ²⁵
FGaT/MoSe ₂ /FGaT	6.1 nm MoSe ₂	3 % OOP (300 K) 37.7 % OOP (2 K)	Yin et al. ²⁶

- The authors should also clearly state which of the SOT switching schemes is observed and/or characterized: Type-X, Type-Y or Type-Z? More results like those in Fig. 3 but with the external field varied in the three different principal axial directions should be included (perhaps in the supplementary so that the main manuscript is not too long).

Response: Given that the ferromagnetic material in our devices, FGaT, exhibits perpendicular magnetic anisotropy favoring an out-of-plane steady state magnetization ($\vec{m} \parallel \hat{z}$), and that the spin accumulation and spin current from the adjacent heavy metal responsible for the SOT switching is oriented in-plane and orthogonal to the current, our system exhibits the type-z SOT switching scheme. We have also included language in the main text of our revised manuscript to make this categorization explicit. We thank the Reviewer for leading us to this discussion.

In accordance with the Reviewer's directions, we have also performed experiments with one of our new devices, where the current pulses are applied similar to the switching experiments, while the external field is applied along sample normal ($H \parallel c$) and in-plane but orthogonal to the current ($H \parallel b$), in addition to the conventional configuration of ($H \parallel I \parallel a$). These results are

presented in Fig. 6 below and included in the Supplementary Information Section 2 of our revised manuscript.

Figure 6: (a) Current-induced switching loops for the device under increasing external magnetic field applied parallel to the current ($H \parallel I \parallel a$). (b) Variation of the switched anomalous Hall resistance (R_{xy}^{AHE}) with externally applied field, $H \parallel I \parallel a$. (c) Response of the device D2 to current pulsing when the external field is applied out-of-plane ($H \parallel c$). Inset: Enlarged view of the data at high resistances. Under 0 Oe field, we observe a small drop in R_{xy} at maximum current corresponding to a slight demagnetization. For field ≥ 100 Oe, no such demagnetization is observed, as the OOP field keeps the magnetization aligned OOP (d) Response of D2 to current pulsing when the external field is applied in-plane and orthogonal to the current injection direction, i.e. $H \parallel b$. In this case, the field neither assists the deterministic SOT switching, nor does it help saturate the magnetization OOP. Thus, the device which is initially saturated along +z direction, gradually demagnetizes under subsequent pulsing cycles. No clear effect of field variation could be

observed. Measurements were done sequentially under 0, 100, 500, and 1000 Oe respectively. All measurements were performed at 300 K.

For the case of external field applied parallel to the current ($H \parallel I \parallel a$), we observe clear switching loops starting 100 Oe field (Fig. 6a). The variation of switched anomalous Hall resistance with field magnitude shows a downward trend (Fig. 6b), quite similar to that observed for the device D1 reported in the main text. This is expected as increasing the strength of in-plane magnetic field tilts the steady state FGaT magnetization further away from the c -axis, resulting in reduction of the observed anomalous Hall resistance. However, we see a stark difference in the device's response to a cyclic current pulse train when the external field is applied along the other principal axes. As seen in Fig. 6c, under zero external field, the device exhibits a small drop in R_{xy} in response to the current pulses, indicating a partial demagnetization. However, when the current pulses are applied while the external out-of-plane field is ≥ 100 Oe, we do not observe any demagnetization and the R_{xy} value remains close to its saturation level as the out-of-plane (OOP) field helps keep the magnetization saturated. Finally, when the field is applied in-plane and orthogonal to the current ($H \parallel b$) as shown in Fig. 6d, we see a step by step drop in R_{xy} from its saturated value, upon sequential excitation with current pulses. This indicates gradual demagnetization of the FGaT layer since the external field neither helps saturate the magnetization OOP nor does it assist the deterministic switching of the FGaT magnetization. We would like to note that for this case, the device was initialized to OOP magnetization only before loading into the cryostat and could not be initialized again between the measurements as field could only be applied in-plane with that carrier PCB. Thus, we observe a step-by-step demagnetization as the field is varied sequentially from 0 to 1000 Oe. No clear effect of the field magnitude variation could be observed.

- As for the estimation of the spin-torque efficiency of the anti-damping torque, the authors can perhaps refer to the works done by C. L. Chien at John Hopkins University. One of the important findings is that the estimated spin-torque efficiency can be strongly dependent on the measurement and estimation methods used. However, for a fixed method, measurements on different material systems give a fairly good estimate of the relative differences in the spin-torque efficiency factor. This should help to better clarify how accurate the measured spin-torque efficiency is.

Response: The Reviewer has rightly pointed out that the estimation of SOT efficiency is significantly affected by the measurement method used. Given the wide variety of methods that can be found in literature for measuring SOT efficiencies¹⁹, like spin-torque ferromagnetic resonance (ST-FMR), harmonic Hall voltage response (HHVR), inverse spin-Hall effect (ISHE), hysteresis loop shift and spin Hall magnetoresistance, with varying sources of errors and modes for their resolution, it becomes difficult to compare the ξ estimates for different material systems made using different measurement techniques. This nuance was carefully taken into account by us when designing our experiments. Our choice of the HHVR technique is driven by the theoretical

soundness of its estimation method, which rigorously accounts for all spurious contributions from magneto-thermal effects (as we have detailed in previous responses), as well as the fact that it has widely been used not only for bulk FM-based SOT systems, but also in the recently growing domain of vdW FM-based SOT systems. In Table 3 below, we list all the recently reported vdW FM-based SOT switching systems where SOT efficiency is measured. It is clear that the HHVR technique is a method of choice for such systems and provides a reliable way of benchmarking ξ_{DL} for new material systems like ours.

Table 3: ξ_{DL} estimation in vdW FM based SOT systems.

Ferro-magnet	Spin-Hall material	ξ_{DL} estimation method	ξ_{DL}	Reference
Fe ₃ GeTe ₂	Pt	HHVR	0.14	Alghamdi et al. ¹²
Fe ₃ GeTe ₂	Pt	HHVR	0.12	Wang et al. ²⁷
Fe ₃ GeTe ₂	WTe ₂	Hysteresis loop shift	4.6	Shin et al. ²⁸
Cr ₂ Ge ₂ Te ₆	Pt	HHVR	0.25	Gupta et al. ¹¹
Fe ₃ GaTe ₂	Pt	HHVR	0.22†	Li et al. ²⁹ ‡ (unpublished)
Fe ₃ GaTe ₂	Pt	HHVR	0.093	This work.

†Likely overestimation as contributions from thermal effects ignored in calculation of ξ_{DL} .

‡An unpublished work that appeared while we were preparing our manuscript. Our work was done independently.

References

- [1] Gong, C., Li, L., Li, Z., Ji, H., Stern, A., Xia, Y., Cao, T., Bao, W., Wang, C., Wang, Y., Qiu, Z. Q., Cava, R. J., Louie, S. G., Xia, J. & Zhang, X. Discovery of intrinsic ferromagnetism in two-dimensional van der Waals crystals. *Nature* **546**, 265–269 (2017).
- [2] Huang, B., Clark, G., Navarro-Moratalla, E., Klein, D. R., Cheng, R., Seyler, K. L., Zhong, Di., Schmidgall, E., McGuire, M. A., Cobden, D. H., Yao, W., Xiao, D., Jarillo-Herrero, P. & Xu, X. Layer-dependent ferromagnetism in a van der Waals crystal down to the monolayer limit. *Nature* **546**, 270–273 (2017).
- [3] Fei, Z., Huang, B., Malinowski, P., Wang, W., Song, T., Sanchez, J., Yao, W., Xiao, D., Zhu, X., May, A. F., Wu, W., Cobden, D. H., Chu, J.-H. & Xu, X. Two-dimensional itinerant ferromagnetism in atomically thin Fe₃GeTe₂. *Nat. Mater.* **17**, 778–782 (2018).
- [4] Seo, J., Kim, D. Y., An, E. S., Kim, K., Kim, G.-Y., Hwang, S.-Y., Kim, D. W., Jang, B. G., Kim, H., Eom, G., Seo, S. Y., Stania, R., Muntwiler, M., Lee, J., Watanabe, K., Taniguchi, T., Jo, Y. J., Lee, J., Min, B. Il, Jo, M. H., Yeom, H. W., Choi, S.-Y., Shim, J. H. & Kim, J. S. Nearly room temperature ferromagnetism in a magnetic metal-rich van der Waals metal. *Sci. Adv.* **6**, eaay8912 (2020).
- [5] May, A. F., Ovchinnikov, D., Zheng, Q., Hermann, R., Calder, S., Huang, B., Fei, Z., Liu, Y., Xu, X. & McGuire, M. A. Ferromagnetism Near Room Temperature in the Cleavable van der Waals Crystal Fe₅GeTe₂. *ACS Nano* **13**, 4436–4442 (2019).
- [6] Zhang, G., Guo, F., Wu, H., Wen, X., Yang, L., Jin, W., Zhang, W. & Chang, H. Above-room-temperature strong intrinsic ferromagnetism in 2D van der Waals Fe₃GaTe₂ with large perpendicular magnetic anisotropy. *Nat. Commun.* **13**, 1–8 (2022).
- [7] Li, J., Yu, G., Tang, C., Liu, Y., Shi, Z., Liu, Y., Navabi, A., Aldosary, M., Shao, Q., Wang, K. L., Lake, R. & Shi, J. Deficiency of the bulk spin Hall effect model for spin-orbit torques in magnetic-insulator/heavy-metal heterostructures. *Phys. Rev. B* **95**, 1–5 (2017).
- [8] Husain, S., Figueiredo-Prestes, N., Fayet, O., Collin, S., Godel, F., Jacquet, E., Reyren, N., Jaffrès, H. & George, J. M. Origin of the anomalous Hall effect at the magnetic insulator/heavy metals interface. *Appl. Phys. Lett.* **122**, (2023).
- [9] Dutta, S., Bose, A., Tulapurkar, A. A., Buhrman, R. A. & Ralph, D. C. Interfacial and bulk spin Hall contributions to fieldlike spin-orbit torque generated by iridium. *Phys. Rev. B* **103**, 1–7 (2021).
- [10] Cha, I. H., Lee, M. H., Kim, G. W., Kim, T. & Kim, Y. K. Spin-orbit torque efficiency in Ta or W/Ta-W/CoFeB junctions. *Mater. Res. Express* **8**, (2021).
- [11] Gupta, V., Cham, T. M., Stiehl, G. M., Bose, A., Mittelstaedt, J. A., Kang, K., Jiang, S., Mak, K. F., Shan, J., Buhrman, R. A. & Ralph, D. C. Manipulation of the van der Waals Magnet Cr₂Ge₂Te₆ by Spin-Orbit Torques. *Nano Lett.* **20**, 7482–7488 (2020).
- [12] Alghamdi, M., Lohmann, M., Li, J., Jothi, P. R., Shao, Q., Aldosary, M., Su, T., Fokwa, B. P.

- T. & Shi, J. Highly Efficient Spin-Orbit Torque and Switching of Layered Ferromagnet Fe₃GeTe₂. *Nano Lett.* **19**, 4400–4405 (2019).
- [13] Avci, C. O., Garello, K., Gabureac, M., Ghosh, A., Fuhrer, A., Alvarado, S. F. & Gambardella, P. Interplay of spin-orbit torque and thermoelectric effects in ferromagnet/normal-metal bilayers. *Phys. Rev. B - Condens. Matter Mater. Phys.* **90**, 1–11 (2014).
- [14] Roschewsky, N., Walker, E. S., Gowtham, P., Muschinske, S., Hellman, F., Bank, S. R. & Salahuddin, S. Spin-orbit torque and Nernst effect in Bi-Sb/Co heterostructures. *Phys. Rev. B* **99**, 1–5 (2019).
- [15] Zhang, K., Lee, Y., Coak, M. J., Kim, J., Son, S., Hwang, I., Ko, D. S., Oh, Y., Jeon, I., Kim, D., Zeng, C., Lee, H. W. & Park, J. G. Highly Efficient Nonvolatile Magnetization Switching and Multi-Level States by Current in Single Van der Waals Topological Ferromagnet Fe₃GeTe₂. *Adv. Funct. Mater.* **31**, (2021).
- [16] Johansen, Ø., Risinggård, V., Sudbø, A., Linder, J. & Brataas, A. Current Control of Magnetism in Two-Dimensional Fe₃GeTe₂. *Phys. Rev. Lett.* **122**, 1–6 (2019).
- [17] Zhang, K., Han, S., Lee, Y., Coak, M. J., Kim, J., Hwang, I., Son, S., Shin, J., Lim, M., Jo, D., Kim, K., Kim, D., Lee, H. W. & Park, J. G. Gigantic Current Control of Coercive Field and Magnetic Memory Based on Nanometer-Thin Ferromagnetic van der Waals Fe₃GeTe₂. *Adv. Mater.* **33**, 1–8 (2021).
- [18] Miron, I. M., Garello, K., Gaudin, G., Zermatten, P. J., Costache, M. V., Auffret, S., Bandiera, S., Rodmacq, B., Schuhl, A. & Gambardella, P. Perpendicular switching of a single ferromagnetic layer induced by in-plane current injection. *Nature* **476**, 189–193 (2011).
- [19] Nguyen, M.-H. & Pai, C.-F. Spin-orbit torque characterization in a nutshell. *APL Mater.* **9**, 30902 (2021).
- [20] Kim, J., Sinha, J., Hayashi, M., Yamanouchi, M., Fukami, S., Suzuki, T., Mitani, S. & Ohno, H. Layer thickness dependence of the current-induced effective field vector in Ta|CoFeB|MgO. *Nat. Mater.* **12**, 240–245 (2013).
- [21] Garello, K., Miron, I. M., Avci, C. O., Freimuth, F., Mokrousov, Y., Blügel, S., Auffret, S., Bouille, O., Gaudin, G. & Gambardella, P. Symmetry and magnitude of spin-orbit torques in ferromagnetic heterostructures. *Nat. Nanotechnol.* **8**, 587–593 (2013).
- [22] Peng, S., Wang, M., Yang, H., Zeng, L., Nan, J., Zhou, J., Zhang, Y., Hallal, A., Chshiev, M., Wang, K. L., Zhang, Q. & Zhao, W. Origin of interfacial perpendicular magnetic anisotropy in MgO/CoFe/metallic capping layer structures. *Sci. Rep.* **5**, 18173 (2015).
- [23] Zhu, W., Xie, S., Lin, H., Zhang, G., Wu, H., Hu, T., Wang, Z., Zhang, X., Xu, J., Wang, Y., Zheng, Y., Yan, F., Zhang, J., Zhao, L., Patané, A., Zhang, J., Chang, H. & Wang, K. Large Room-Temperature Magnetoresistance in van der Waals Ferromagnet/Semiconductor Junctions. *Chinese Phys. Lett.* **39**, (2022).

- [24] Jin, W., Zhang, G., Wu, H., Yang, L., Zhang, W. & Chang, H. Room-Temperature and Tunable Tunneling Magnetoresistance in Fe₃GaTe₂-Based 2D van der Waals Heterojunctions. *ACS Appl. Mater. Interfaces* **15**, 36519–36526 (2023).
- [25] Jin, W., Zhang, G., Wu, H., Yang, L., Zhang, W. & Chang, H. Room-temperature spin-valve devices based on Fe₃GaTe₂/MoS₂/Fe₃GaTe₂ 2D van der Waals heterojunctions. *Nanoscale* **15**, 5371–5378 (2023).
- [26] Yin, H., Zhang, P., Jin, W., Di, B., Wu, H., Zhang, G., Zhang, W. & Chang, H. Fe₃GaTe₂/MoSe₂ ferromagnet/semiconductor 2D van der Waals heterojunction for room-temperature spin-valve devices. *CrystEngComm* **25**, 1339–1346 (2023).
- [27] Wang, X., Tang, J., Xia, X., He, C., Zhang, J., Liu, Y., Wan, C., Fang, C., Guo, C., Yang, W., Guang, Y., Zhang, X., Xu, H., Wei, J., Liao, M., Lu, X., Feng, J., Li, X., Peng, Y., Wei, H., Yang, R., Shi, D., Zhang, X., Han, Z., Zhang, Z., Zhang, G., Yu, G. & Han, X. Current-driven magnetization switching in a van der Waals ferromagnet Fe₃GeTe₂. *Sci. Adv.* **5**, 2–8 (2019).
- [28] Shin, I., Cho, W. J., An, E. S., Park, S., Jeong, H. W., Jang, S., Baek, W. J., Park, S. Y., Yang, D. H., Seo, J. H., Kim, G. Y., Ali, M. N., Choi, S. Y., Lee, H. W., Kim, J. S., Kim, S. D. & Lee, G. H. Spin–Orbit Torque Switching in an All-Van der Waals Heterostructure. *Adv. Mater.* **34**, 1–7 (2022).
- [29] Li, W., Zhu, W., Zhang, G., Wu, H., Zhu, S., Li, R., Zhang, E., Zhang, X., Deng, Y., Zhang, J., Zhao, L., Chang, H. & Wang, K. Room-temperature van der Waals 2D ferromagnet switching by spin-orbit torques. *arXiv* **2304.10718**, (2023).

Reviewers' Comments:

Reviewer #1:

Remarks to the Author:

It is nice to receive the author's reply and revisions. Having thoroughly read the reply letter and also the revisions, I somehow feel I might appear to give too much suggestion during the previous review (I actually also spent a long time to finish reading all these replies).

As I mentioned in my last review, I would be happy to see this interesting work accepted in the vdW magnet and spintronic field. My former position was to strengthen the paper, so I raised 10 comments in details but I didn't expect the authors to address all of them. Surprisingly, all the comments have been addressed thoroughly and faithfully, and the manuscript has also been significantly improved. I really appreciate the author's enthusiasm to research and also the logics they used in their rebuttal letter. Well done! Therefore, I would recommend its publication at this time without any further change.

Reviewer #2:

Remarks to the Author:

The authors have replied to the comments of the reviewers with supporting data and remarks. I feel this is sufficient for publication in nature communications.

Reviewer #3:

Remarks to the Author:

I am satisfied with the revisions done to the manuscript and the authors have sufficiently addressed my concerns.